# Polarisome scaffolder Spa2-mediated macromolecular condensation of Aip5 for actin polymerization

Ying Xie [1,5], Jialin Sun[2,3,5], Xiao Han[3], Alma Turšić-Wunder[3], Joel D.W. Toh[2,3], Wanjin Hong [2], Yong-Gui Gao[2,3,4,6]* & Yansong Miao [1,3,6]*

A multiprotein complex polarisome nucleates actin cables for polarized cell growth in budding yeast and filamentous fungi. However, the dynamic regulations of polarisome proteins in polymerizing actin under physiological and stress conditions remains unknown. We identify a previously functionally unknown polarisome member, actin-interacting-protein 5 (Aip5), which promotes actin assembly synergistically with formin Bni1. Aip5-C terminus is responsible for its activities by interacting with G-actin and Bni1. Through N-terminal intrinsically disordered region, Aip5 forms high-order oligomers and generate cytoplasmic condensates under the stresses conditions. The molecular dynamics and reversibility of Aip5 condensates are regulated by scaffolding protein Spa2 via liquid-liquid phase separation both in vitro and in vivo. In the absence of Spa2, Aip5 condensates hamper cell growth and actin cable structures under stress treatment. The present study reveals the mechanisms of actin assembly for polarity establishment and the adaptation in stress conditions to protect actin assembly by protein phase separation.

[1] School of Chemical and Biomedical Engineering, Nanyang Technological University, Singapore 637459, Singapore. [2] Institute of Molecular and Cell Biology, A*STAR, Singapore 138673, Singapore. [3] School of Biological Sciences, Nanyang Technological University, Singapore 637551, Singapore. [4] NTU Institute of Structural Biology, Nanyang Technological University, 59 Nanyang Drive, Singapore 639798, Singapore. [5]These authors contributed equally: Ying Xie, Jialin Sun. [6]These authors jointly supervised this work: Yong-Gui Gao, Yansong Miao. *email: ygao@ntu.edu.sg; yansongm@ntu.edu.sg

The polarized growth of both yeast and filamentous fungi is primarily driven by actin polymerization that is initiated from the polarisome protein complex at the elongating tip[1–4]. In the budding yeast, the polarisome complex is an actin cable nucleation center that consists of the following core members: formin Bni1, nucleation-promoting factor (NPF) Bud6, Pea2, and scaffolding protein Spa5[5,6]. Actin assembly acutely responds to environmental stimuli, which triggers various signal transduction mechanisms[7–9]. Polarisome complex proteins are intrinsically disordered proteins (IDPs) that have a high ratio of disordered residues to integrate interior and exterior signals by changing protein conformations or protein–protein interactions[10]. Polarisome proteins are located in a non-membranous zone that changes its size and protein density at the bud tip during polarized cell growth when cells switch from apical to isotropic growth[11–13].

The Arp2/3 complex, the formins, and WASP homology 2 (WH2) are three major classes of actin nucleation factors (NFs)[14,15]. The nucleation activities of NF, Arp2/3 complex, or formins can be regulated by NPF or another NF[14]. For example, the interaction between WH2 class-NF Spire and formin class-NF Fmn2, or adenomatous polyposis coli protein and mDia1, creates synergy in actin nucleation by directly activating the formin[15–17]. Furthermore, recent studies have shown that multivalent protein–protein interactions play essential roles in the liquid phase separation that spatiotemporally regulates diverse biological reactions. Multivalent interactions with Nephrin and Nck activates N-WASP in actin nucleation via phase separation[9,18,19]. Studies of the protein–protein interaction or protein multivalence of actin nucleators and NPFs offer essential implications in understanding the in vivo complex assembly of different types of actin networks within the same crowded cytoplasm. In the budding yeast, the molecular mechanisms by which polarisome members concentrate in the non-membranous compartment to cooperatively regulate actin assembly are poorly understood, especially under different physiological or stress cues.

Here we report a previously unknown budding yeast polarisome protein encoded by *YFR016C*, named actin-interacting protein 5 (Aip5). Aip5 is found to accelerate the formation of actin filaments in vitro through its C-terminal folded-domain (Aip5-C) and mediates actin cable assembly in cells. Structural studies by X-ray crystallography and biochemical assays have identified that Aip5-C is in a dimeric form, in which the dimer state and a loop region that directly binds to G-actin play essential roles in regulating actin assembly activity. Strikingly, Aip5 also serves to enhance Bni1 nucleation activities by its direct interaction with Bni1 C-terminus. To maintain the local protein concentration at the elongating tip, the N-terminal intrinsically disordered region (IDR) of Aip5 confers unique oligomerization properties, which also directly interacts with the C-terminus of polarisome scaffolding protein Spa2. Spa2 undergoes liquid–liquid phase separation (LLPS) that recruits and converts the Aip5 from amorphous assemblies into liquid states under crowding environment. In vivo, physiological stresses of energy depletion and changing of intracellular pH induce reversible Aip5 aggregates, in a Spa2-dependent manner, indicating an adaptation mechanism to rescue Aip5 proteins and dissolve their assemblies rapidly during stress recovery, and thus ensuring a restart of actin assembly.

## Results

### Aip5 and Bni1 synergistically promote actin assembly.
In prior Bni1-based reconstitution of the actin cable assembly, we identified a functionally unknown protein, encoded by *YFR016C*[20], named Aip5. Aip5 is conserved among fungal species, including various filamentous fungi that are pathogenic for humans and plants (Supplementary Fig. 1a). Aip5 has been detected by an immunoprecipitation assay using the polarisome scaffolding protein Spa2[4]. To understand the Aip5 relationship with polarisome protein complex, we first determined Aip5 in vivo localization using a C-terminal green fluorescent protein (GFP) tagging at the endogenous locus. It showed that Aip5 localized to the polarized growing tip and bud neck, entirely dependent on the presence of Spa2 (Fig. 1a). PHYRE2 and ANCHOR predictions showed that Spa2 was primarily intrinsically disordered at the N-terminus except for the C-terminal region (residues 1306–1466aa, Spa2-C) (Supplementary Fig. 1b). Therefore, we generated a Spa2 C-terminal truncation mutant, *spa2ΔC* (1–1305aa), in which the concentrated tip and neck localization of Aip5 was abolished, though *spa2ΔC*-GFP still localized in vivo as wild-type (WT) Spa2 (Fig. 1a, b and Supplementary Fig. 1c). At the growing tip, Aip5 co-localized with polarisome members Bni1 and Bud6, in which the colocalization was more pronounced in the punctate foci upon Latrunculin A (LatA) treatment (Supplementary Fig. 1c). Aip5 polarity was also partially dependent on Bni1 and Bud6, given that both *bni1Δ* and *bud6Δ* showed an attenuated Aip5-GFP localization (Fig. 1a, b). The localization change of Aip5 in the polarisome mutant was unlikely to be caused by Aip5 degradation (Supplementary Fig. 1d). Furthermore, actin depolymerization by LatA concentrated Bni1 from the cytoplasm to bright dots at the bud tip[21], where Aip5 showed similar tip enrichment that is partially remained in *spa2Δ* but disappeared entirely in *spa2Δ bni1Δ* (Supplementary Fig. 1e, f), suggesting potential direct interactions between Bni1 and Aip5. We were motivated to explore whether Aip5 has similar activities in promoting actin assembly as the other polarisome members, Bni1 and Bud6[22]. Aip5 that was purified from yeast and showed to enhance spontaneous actin filament assembly in a dose-dependent manner from the submicromolar to the micromolar range (Fig. 1c, d). Aip5 promoted actin assembly was not due to F-actin severing that could produce more barbed ends and thus increased the polymerization rate[23] (Supplementary Fig. 1g, h). Surprisingly, through real-time monitoring actin polymerization using total internal reflection fluorescence microscopy (TIRFM) assay, Aip5 also showed a synergistic effect with Bni1FH1COOH in promoting actin nucleation, by generating a significant number of short actin filaments (Fig. 1e–g, Supplementary Fig. 1i, j) but not through increasing of barbed-end elongation (Fig. 1h, i). Together, our results suggested Aip5 promotes actin assembly on its own and synergistically with formin Bni1 to increase actin nucleation in vitro (Fig. 1d–g), indicating its potential function as NPF of Bni1.

To further investigate the synergy between Bni1 and Aip5 in actin assembly, we next examined the in vivo Aip5 function using yeast mutants *aip5Δ* in the background of *bni1Δ* and the ts-allele *bni1-12 bnr1Δ* and monitored in cell growth and actin cable assembly. Both *aip5Δ bni1Δ* and *aip5Δ bni1-12 bnr1Δ* demonstrated apparent synthetic sickness, compared to the same background with *AIP5*, suggesting the negative genetic interaction between *BNI1* and *AIP5*. (Fig. 1j and Supplementary Fig. 2a). Subsequently, we tested the cell growth rate of WT and mutants under the conditions of compromised actin assembly by applying a low dose of LatA. While *aip5Δ* showed higher sensitivity than WT to the actin perturbation, the LatA treatment causes strong growth inhibition of the *aip5Δ bni1Δ* (Fig. 1k). We next investigated how Aip5 regulates actin polymerization in vivo using actin cable marker Abp140-3xGFP. The mutant *aip5Δ bni1Δ* displayed a significant defect in actin cable assembly both in the cable length and the detectable Abp140 signal intensity per actin cable (Supplementary Fig. 2b–d) and a moderate decrease of actin cable number per cell (Supplementary Fig. 2e). Consistently,

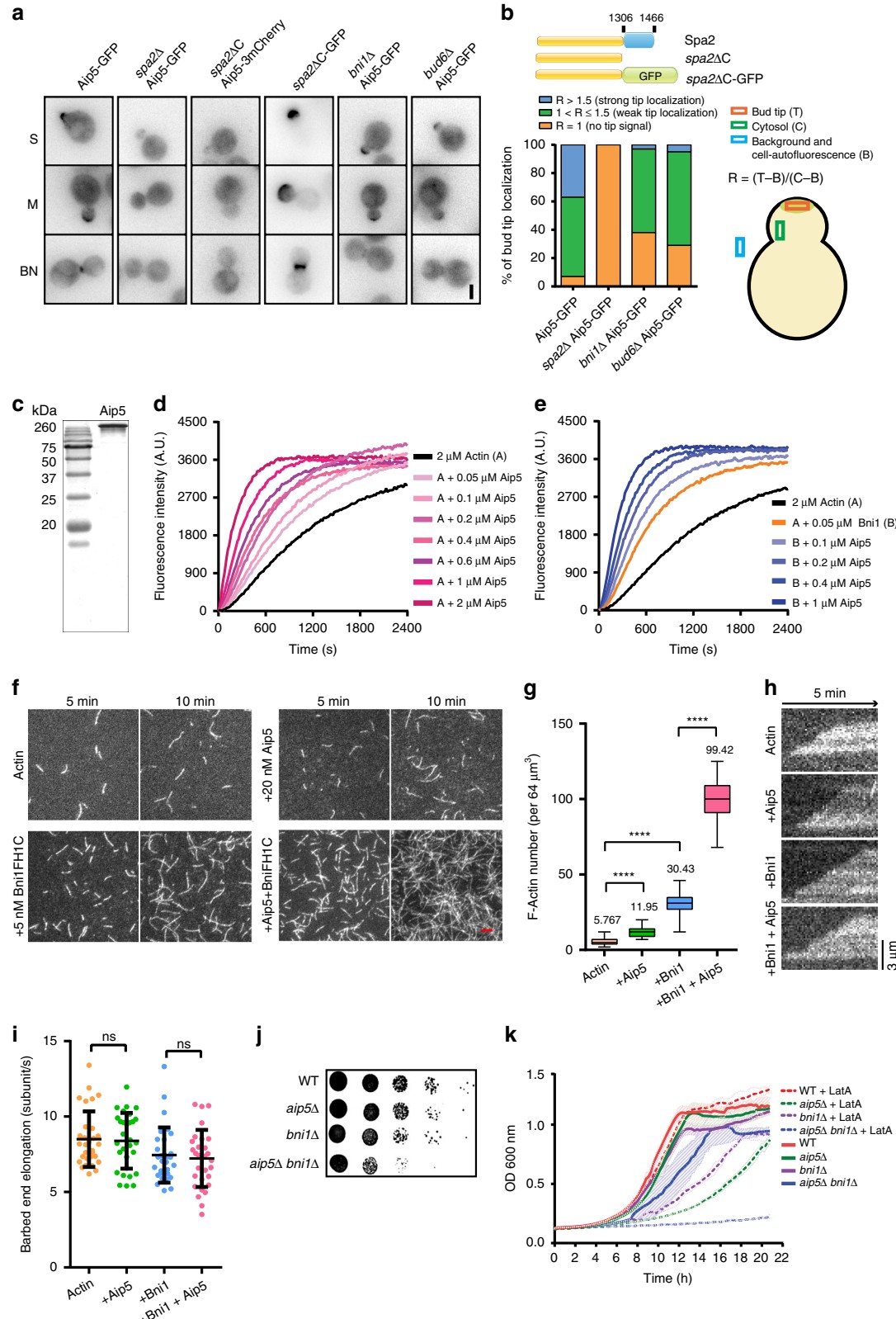

without *BNR1*, *aip5Δ bni1-12 bnr1Δ* also significantly reduced the actin cable number, compared to *bni1-12 bnr1Δ* (Supplementary Fig. 2f, g). In contrast, Aip5 function in actin assembly seems to be actin cable specific, because there were no evident defects of endocytosis efficiency in *aip5Δ* or *aip5Δ bni1Δ* (Supplementary Fig. 2h, i).

**Aip5 C-terminus promotes Bni1-mediated actin assembly.** We next sought to identify the functional domain of Aip5 in promoting actin assembly. The structural prediction suggests that Aip5 (1–1131aa) is mostly unstructured with low amino acid complexity (Fig. 2a), while the short C-terminus is a well-ordered domain (1132–1234aa). Therefore, we designed several constructs

**Fig. 1** Aip5 accelerates bulk actin assembly and Bni1-mediated actin polymerization in the polarisome. **a** Representative maximum Z-projection images of Aip5-GFP at the presumptive bud site in wild-type and polarisome yeast mutants. S small-budded cell, M medium size-budded cell, BN bud neck. Scale bar represents 3 μm. **b** Domain schematic of the polarisome scaffold protein Spa2. Three classes of Aip5 localization that are quantitatively determined by the ratio of fluorescence signal intensity (R) of Aip5-GFP at presumptive bud site and cytoplasm (Aip5-GFP, $n = 41$ bud sites; $spa2\Delta$ Aip5-GFP, $n = 23$ bud sites; $bni1\Delta$ Aip5-GFP, $n = 29$ bud sites; $bud6\Delta$ Aip5-GFP, $n = 59$ bud sites). **c** The SDS-PAGE gel of purified Aip5 protein from budding yeast. **d** Pyrene-actin polymerization reaction with an increased concentration of full-length Aip5. **e** Pyrene-actin polymerization reaction of Aip5 in the presence of Bni1FH1COOH. **f** Representative total internal reflection fluorescence microscope (TIRFM) images of actin filaments formed at 5 and 10 min in the field of 48 μm³. Actin filaments were assembled using 0.5 μM actin (10% Oregon green 488 labeled and 0.5% biotin–actin), with or without 5 nM Bni1FH1COOH and 20 nM Aip5. The scale bar represents 5 μm. **g** Quantification of the number of nucleated-actin seeds at 5 min ($n = 60$ ROIs for each sample, ROI = 64 μm³). The box plot covers data from minimal to maximal with the central line indicating the mean value. **h** Representative kymograph of elongating actin filament generated from movies over 5 min with 5-s interval between each frame. The scale bar represents 3 μm. **i** Quantification of actin filament barbed-end elongation speed of the indicated protein combinations as shown in **f** ($n = 30$ actin filament for each sample). **j** Yeast spotting assay of the indicated strains that were grown on YPD plate at 25 °C for 36 h before imaging. **k** Yeast growth curves of the liquid cultures at 25 °C in the presence of 1 μM LatA ($n = 4$). p Values were determined by one-way ANOVA, ns not significant, ****$p < 0.0001$. All the bar graph represents mean value; error bar, S.D. Source data are provided as a Source Data file

containing the C-terminal region and identified that one version spanning residues 1110–1233 (referred to as Aip5-C) displayed high solubility (Fig. 2b, c). Aip5-C directly interacts with G-actin at a moderate affinity, with a $K_D$ value of ~$1.5 \pm 0.9$ μM ($n = 3$ with S.D.) (Fig. 2d). Aip5-C per se exhibited actin filament assembly activity starting from tens of nanomolar to micromolar using pyrene-actin polymerization experiment (Fig. 2e). Under TIRFM actin assembly assays, Aip5-C directly enhanced Bni1-mediated nucleation activity by increasing the amount of short F-actin seeds (Fig. 2f, g). Unlike the Bni1[24], Aip5-C could not reverse the CapZ-blocked actin elongation independently (Fig. 2h) but showed an additive effect in enhancing the Bni1FH1COOH activity in releasing the CapZ from the barbed-end (Fig. 2h). Furthermore, we sought to investigate the mechanisms by which Aip5-C promotes Bni1-mediated actin assembly. We purified the recombinant protein of several Bni1 truncating variants by dividing the Bni1FH1COOH into three distinct domains, FH1, FH2, and C-terminus, in which both FH1 and C-terminus were predicted to be intrinsically disordered (Fig. 2i–k). We first tested the Aip5-C and Bni1 interaction by the fluorescence anisotropy (FA) binding assay. Bni1-C directly binds to Aip5-C with a high affinity ($K_D = 110 \pm 106$ nM). Bni1FH2 does not show any interaction with Aip5-C, although Bni1FH2C ($K_D = 36 \pm 15$ nM) that contains both FH2 and C regions had approximately threefold higher affinity compared to Bni1-C alone (Fig. 2l). An additional FH1 domain confers Bni1FH1COOH an increase in affinity to a $K_D = 16 \pm 10$ nM (Fig. 2l). It is possible that the adjacent domain of Bni1-C enhanced the stability when it interacted with Aip5-C and thus increasing the binding affinity. We could not obtain soluble recombinant protein of FH1 and were therefore unable to determine the affinity between FH1 and Aip5. We then examined as to how FH1 and C-terminus contribute to the synergy between Aip5-C and Bni1, respectively. Pyrene-actin-based assembly rates were measured using an increasing concentration of Bni1 variants, in the presence or absence of Aip5-C. The actin polymerization of either Bni1FH1COOH or Bni1FH2C was synergistically promoted by Aip5-C (Fig. 2m). However, Aip5-C could not enhance the actin assembly rate for Bni1FH1FH2 that lacks Bni1-C (Fig. 2m), indicating a Bni1-C-dependent promoting activity of Aip5.

**A stable loop region of Aip5-C interacts with G-actin.** Aip5-C forms dimer in solution, in which the estimated molecular weight of Aip5-C is around 15 kDa, whereas the calculated molecular weight is 31.7 kDa (Fig. 3a). Given the actin polymerization activities, we sought to elucidate the molecular mechanism of Aip5-C function by structural studies using X-ray crystallography. The crystals obtained diffraction beyond 2.0 Å. The

structure of Aip5-C was determined by the selenomethionine-multiple wavelength anomalous diffraction (SeMet-MAD) phasing method (Table 1). The crystalized Aip5-C forms a homo-dimer, in which protomers A and B consist of four β-sheets in the core region flanked by four α-helices (Fig. 3b). This structural arrangement can be subdivided into an N-terminal β1α1β2 motif and a C-terminal β3β4α3α4 motif, representing a typical thioredoxin fold[25]. Interactions within the dimer interface are predominantly hydrogen bonds that are formed by water molecules (3, 5, 10, 18, 20, 101) and two β2 strands to stabilize the dimer (Fig. 3c). To identify potential functional crosstalk between Aip5-C and thioredoxin fold proteins, we superposed the crystal structure of Aip5-C with an oxidoreductase cGrx2 (PDB ID: 4TR0), which was among the top hits from analysis with DALI server[26]. Interestingly, superposition of Aip5-C with cGrx2 revealed that the substrate glutathione disulfide-binding site (CxxC motif) of cGrx2 is replaced by a hydrophobic loop (SLAGGGFH) connecting β1 and α1 in Aip5-C (Fig. 3d, e and Supplementary Fig. 3a, b). Superposition of protomers A and B showed a root mean square deviation (RMSD) value of 1.5 Å, which indicates high similarity between the two protomers, except for the lack of clear electron density for the loop region between β1 and α1 in protomer B (Fig. 3f). The only seleno-methionine used to solve the structure is located near the loop. Thus the stabilization of the loop between β1 and α1 in Aip5-C is serendipitous for structure determination, which can be seen from the clear electron density in protomer A (Fig. 3d).

The structured loop (SLAGGGFH) in Aip5-C and its absence in cGrx2 motivated us to ask whether the loop contributes to its G-actin recognition and actin assembly activity. We found that the deletion of the loop (Aip5-C-LD) entirely abolished the Aip5 actin assembly activity and its binding to G-actin (Fig. 3g–i and Supplementary Fig. 3c). Using a label-free microscale thermophoresis (MST) assay, we also confirmed the direct binding between G-actin and Aip5 loop, by titrating G-actin with increasing concentrations of a loop containing peptide (TSLAGGGFHM) ($K_D = 99 \pm 76$ μM) (Fig. 3j). L1150 in the loop region is highly conserved among multiple pathogenic filamentous fungi homologs (Supplementary Figs. 1a and 3d). To test the L1150 role in actin filament assembly, we generated a protein mutant Aip5-C-L1150A and additional mutations of the amino acids adjacent to the loop region, single mutants Aip5-C-N1162A and Aip5-C-S1165C, and one triple mutant Aip5-C-3m (L1150A, N1162A, S1165C) (Fig. 3g and Supplementary Fig. 3c). The activities of all Aip5-C variants were compared by calculating the initial polymerization rates using the slope of the secant lines of the first 60–180 s of the normalized data. Of the three single mutants, Aip5-C-L1150A showed the most drastic reduction in

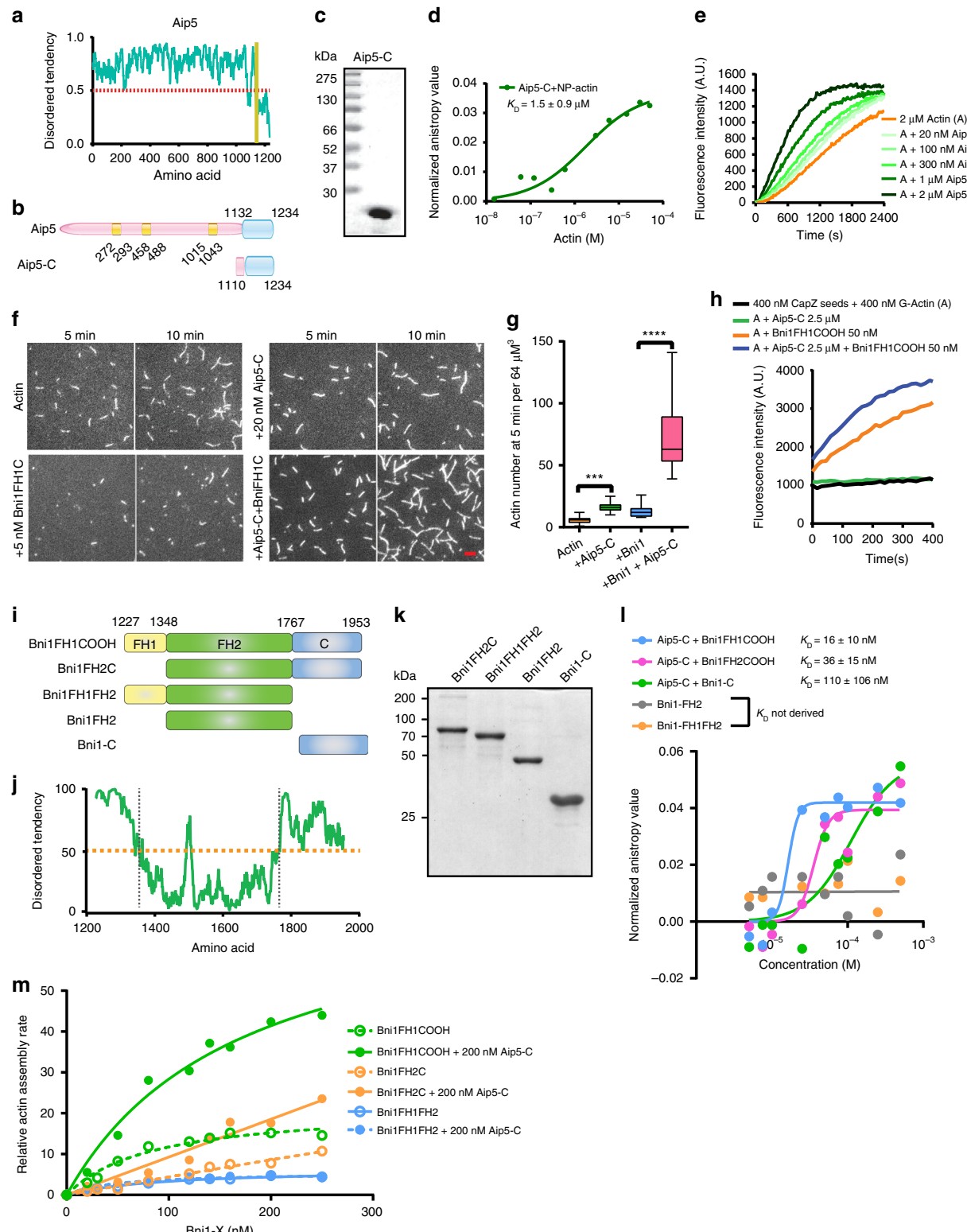

the polymerization rate (Fig. 3k). Notably, the additional S1165C and N1162A mutations to L1150A (Aip5-C-3m) resulted in complete suppression of the actin assembly activity of Aip5-C (Fig. 3k and Supplementary Fig. 3e). To understand the consequences of the triple mutation, Aip5-C-3m was crystallized, and the structure was determined through molecular replacement using the native Aip5-C structure as the search model.

Superposition of protomer A of Aip5-C and Aip5-C-3m structures showed an RMSD of 2.75 Å, which revealed missing electron density for the loop in Aip5-C-3m, indicating increased loop flexibility (Supplementary Fig. 3f), whereas the superposition of protomer B showed an RMSD of 1.88 Å, which showed more similar conformations inside the well-folded region of Aip5-C (Supplementary Fig. 3g). The Aip5-C-3m still formed a

**Fig. 2** Aip5 C-terminus accelerates actin assembly via direct interaction with G-actin and Bni1 C-terminus. **a** IDR prediction of Aip5 using ANCHOR. **b** Domain schematics of Aip5. The yellow regions indicate predicted as coiled-coil domains and the folded C-terminus is in blue. **c** The SDS-PAGE gel of the purified Aip5-C. **d** Fluorescence anisotropy binding measurements of Alexa-488-labeled 30 nM Aip5-C that was titrated by non-polymerized actin (NP-actin). The average values with an error bar of ±S.D. were calculated from three biological replicates and plotted with the Hill slope equation. **e** Pyrene-actin polymerization reaction with an increased concentration of Aip5-C. **f** Representative TIRFM images of actin filaments formed at 5 and 10 min in the same field. Actin filaments were assembled using 0.5 µM actin (10% Oregon green 488 labeled and 0.5% biotin–actin), with or without 5 nM Bni1FH1COOH and 20 nM Aip5. The scale bar represents 5 µm. **g** Quantification of the number of nucleated-actin seeds at 5 min. (control, $n = 45$ ROIs; +Aip5-C, $n = 53$ ROIs; +5 nM Bni1FH1COOH, $n = 45$ ROIs; +Bni1FH1COOH+Aip5-C, $n = 45$ ROIs, ROI = 64 µm$^3$) The box plot covers data from minimal to maximal with the central line indicating the mean value. **h** Barbed-end capping assay using 0.5 µM actin seeds, 10 nM CapZ, and an additional 0.4 µM monomeric pyrene actins, in the presence of Aip5-C with or without Bni1FH1COOH. **i** Domain schematics of Bni1-truncating protein variants, including the different combinations of FH1 domain (1227–1348aa), FH2 (1349–1767aa), and C-terminal region (1768–1953aa). **j** IDR prediction of Bni1FH1COOH using ANCHOR. **k** SDS-PAGE of purified Bni1-truncating protein variants. **l** Anisotropic measurements of Alexa-488-labeled Aip5-C (30 nM) that was titrated with increasing concentration of four truncating Bni1 variants, as in **i**, respectively. The average values with an error bar of ±S.D. were calculated from three biological replicates and plotted with the Hill slope equation. **m** Relative actin assembly rate of the indicated concentration of Bni1 variants in the presence of 200 nM Aip5-C. Data were normalized to the spontaneous actin polymerization. The plot shows average values that were derived from two independent experiments. Source data are provided as a Source Data file

homodimer, however, and displayed significant conformational change when superposed with the native Aip5-C dimer. The loops located in the region of protomers A and B in the Aip5-C dimer run almost parallel with the heads of α1s pointing toward each other (Supplementary Fig. 3h, i). The hydrophobic contact established between two L1150 sites of Aip5-C promoters is, however, absent in the Aip-5-C-3m dimer. Furthermore, the loop sites in both protomers are far apart, with the heads of α1s pointing in opposite directions (Supplementary Fig. 2j). Structural comparisons between Aip5-C and Aip5-C-3m suggest that the flexibility and/or the position of the loop plays a critical role in its actin assembly activities. We also found that the binding to G-actin by the loop is critical for Aip5-C to enhance the Bni1-mediated actin nucleation. The removal of the loop region of Aip5-C completely abolished its activity in promoting Bni1-mediated actin assembly (Fig. 3l).

To further ask the functional importance of dimeric formation in actin assembly, we managed to generate a monomeric form of recombinant Aip5 by removing the 35 amino acids at N-terminal of Aip5-C, namely, Aip5-mC (Supplementary Fig 4a, b). Aip5-mC has eluted from size-exclusion chromatography at around 13 ml, indicating a size of a monomer (~12 kDa) (Supplementary Fig. 4c). Though Aip5-mC was still able to promote actin polymerization, its relative activity on assembling actin filament was lower compared to the Aip5-C dimer (Supplementary Fig. 4d, e). Furthermore, the NPF activity of Aip5-mC for Bni1 was also decreased, compared to Aip5-C (Supplementary Fig. 4f), suggesting the necessity of a dimer formation for offering potent activities.

**IDR mediates Aip5 macromolecular oligomerization in vitro.** IDR is one of the common constituents in generating multivalent interactions and modulate the formation of non-membranous organelles. In response to physiological or pathological cues, molecular condensation of IDPs showed broad implications for intracellular processes, including actin assembly[9,10,18,19,27–31]. The predicted low complexity of Aip5-N motivated us to ask how N-IDR would mediate the behaviors and functions of Aip5. We found that full-length Aip5 eluted around 8–10 ml from Superdex™ 200 Increase column where the estimated protein size was >440 kDa, whereas the predicted Aip5 dimer size is 286 kDa (Supplementary Fig. 5a). The early elution on gel filtration chromatography suggested that Aip5 proteins are in oligomeric states, which was further confirmed by transmission electron microscopy (TEM) where Aip5 showed spherical protein complex with irregular size (Fig. 4a). Though a macromolecule in nature, Aip5 was still soluble and remained in the supernatant after high-speed centrifugation (Supplementary Fig. 5b), suggesting its

soluble hetero-oligomeric states. L-arginine, which is known for regulating protein re-folding and the oligomerization states[32,33], effectively impaired the formation of heterogeneous higher-order Aip5 oligomers, although not into a dimeric form by interprotein interaction of C-terminus (Supplementary Fig. 5a, c). The formation of amorphous Aip5 assemblies depended on protein concentration, poly(ethylene glycol) (PEG) concentration, and PEG size, suggesting its phase-separation behavior (Fig. 4b, c, Supplementary Fig. 5d, e). At a condition (2.5% PEG, 1 µM Aip5) right above the binodal phase boundary, Aip5 grew larger over time significantly, indicating the functional Aip5 oligomer system undergoing phase separation toward thermodynamic coexistence. However, we did not observe assemblies up to 36 h at a condition right below the phase boundary (1% PEG, 1 µM Aip5) (Fig. 4d, e). The examined range of Aip5 concentration from 0.1 to 2 µM in vitro closely resembled the in vivo physiological concentration of Aip5 that are tested by microscopic imaging approach. The average concentration of Aip5 in the cytoplasm and at bud tip is ~83 and ~122 nM, respectively. Consistently, a total protein concentration of ~80 nM was also detected by western blot assay (Supplementary Fig. 5f–i). To understand how the sequence and length of N-terminal IDR of Aip5 determine the Aip5 phase behavior, we examined several Aip5 truncating variants, which were designed based on the predicted IDRs and coiled-coils[34]. By shortening its N-terminus, the in vivo concentration of Aip5 at tip and bud neck decreased, which was not due to the protein degradation (Supplementary Fig. 6a–c). The same sets of recombinant Aip5 protein variants (Aip5-N, Aip5-N1014, Aip5-N457, and Aip5-N271) were also purified from yeast system, which showed consistent electrophoresis pattern of their in vivo counterparts (Supplementary Fig. 6d). Circular dichroism (CD) spectroscopy revealed the intrinsically disordered features for Aip5 and its N-terminal-truncating variants, whereas Aip5-C was well folded (Supplementary Fig. 6e, f). In contrast to Aip5-C, none of the N-terminal variants were capable of promoting actin assembly (Supplementary Fig. 6g). As revealed by TEM image of negative staining, Aip5 N-terminal variants showed up as heterogeneous oligomers with a diameter of a few nanometers, in which the size of the oligomers is Aip5-N length dependent. In the presence of crowded environments, Aip5 N-terminal variants form amorphous assemblies, in which Aip5-N and Aip5-N1014 all showed compact assemblies, whereas Aip5-N457 and Aip5-N271 significantly reduced the size of protein assemblies (Supplementary Fig. 6h).

**Spa2 regulates the molecular condensation of Aip5 in vivo.** In vivo Aip5 localization at the bud tip is Spa2-C-terminus

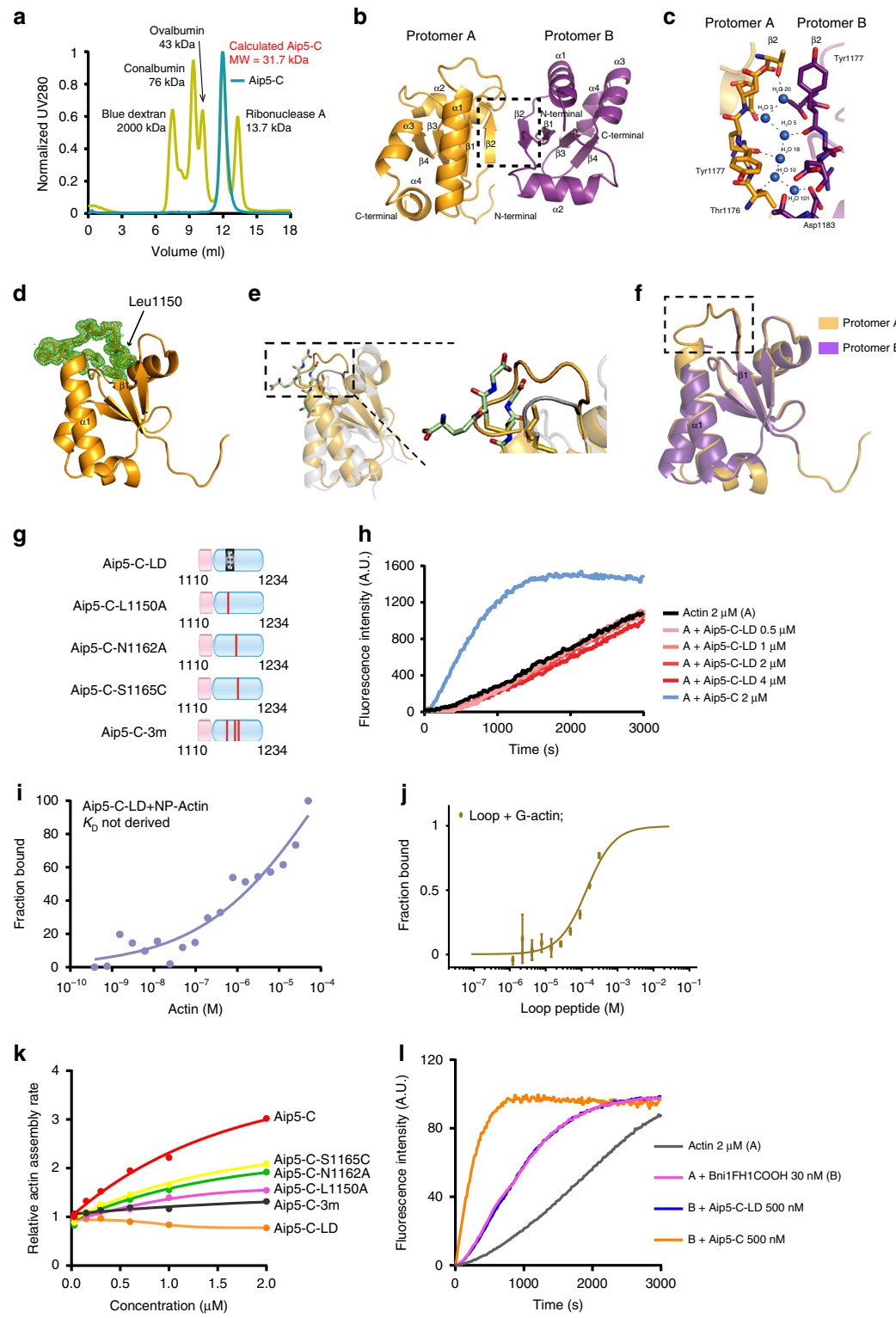

dependent (Fig. 1a), which motivated us to ask how Spa2 regulates the Aip5 oligomerization in vitro and Aip5 phase behavior in vivo. Using surface plasmon resonance (SPR), we proved that the shortest Aip5-N terminal version Aip5-N271 could interact directly with Spa2-C (1306–1466aa), which was in a multivalent binding manner (Fig. 4f and Supplementary Fig. 7a). Furthermore, we generated a recombinant Spa2-LC protein, which

contains predicted disordered-(1151–1305aa) and ordered-(1306–1466aa) regions (Fig. 4g). Under the crowding condition, Spa2-LC but not Spa2-C, formed phase-separated liquid droplets in vitro, indicating the IDR-mediated liquid–liquid phase separation (Fig. 4h, i and Supplementary Fig. 7b, c). Consistent with Spa2–Aip5 interactions, Spa2-LC directly interacted with all four Aip5 N-terminal variants, which showed noticeable slower

**Fig. 3** Aip5-C interacts with G-actin through a stable loop region to deliver actin assembly activity. **a** The elution profile on Aip5-C and protein standards from calibrated-Superdex 75 10/300 GL. The molecular weights of the protein are indicated. Calculated molecular weight of Aip5-C suggested a dimer state in solution. **b** The crystal structure of Aip5-C. The two protomers A and B are colored with bright orange and violet purple, respectively. Dimerization of Aip5-C involves β2 strands from both protomers. **c** An enlarged view of hydrogen bond within the dimer interface of Aip5-C. The β2 strands from both protomers of Aip5-C dimer form hydrogen bonds with water molecules, in particular water molecules 3, 5, 10, 18, 20, and 101 are involved. **d** The electron density map wrapping around the loop region in protomer A of Aip5-C, which reinforces the stability of this loop and assisted in its structural determination. 2Fo-Fc electron density maps are contoured at 1σ. **e** Structural comparison of IxxT motif in Aip5-C with cGrx2 at the corresponding position. **f** Comparison between protomers A and B in Aip5-C dimer. Structures of protomers A and B are vastly similar except for lack of electron density for the loop region in protomer B. **g** Domain schematics of Aip5-C mutation variants. The dashed rectangular box represents a truncated loop region from S1149 to H1156. The red lines represent the mutated residues. **h** Pyrene-actin polymerization by an increasing concentration of Aip5-C-LD. **i** Fluorescence anisotropic binding measurements of Alexa-488-labeled 30 nM Aip5-C-LD that was titrated by non-polymerized actin (NP-actin). Data are represented by circles using the average value of three biological replicates. **j** Microscale thermophoresis binding curves of 2 μM G-actin titrated with peptides of Aip5 loop. Three biological replicates are shown with an error bar of ±S.D. **k** Relative actin assembly rate with the indicated Aip5-C variants after being normalized to the actin-alone reaction. The plot shows the mean value from at least two independent experiments. **l** Pyrene-actin polymerization reaction of Aip5-C and Aip5-C-LD in the presence of Bni1FH1COOH. Source data are provided as a Source Data file

| | **Aip5-C** | **Selenomethionine derivative** | | **Aip5-C-3m** |
|---|---|---|---|---|
| | | **MAD$_{Inflection}$** | **MAD$_{Remote}$** | |
| **Data collection** | | | | |
| Wavelength (Å) | 1.000 | 0.9788 | 0.9636 | 1.5418 |
| Space group | $P 2_1$ | $P 2_1$ | $P 2_1$ | $P 2_1 2_1 2_1$ |
| Cell dimensions | | | | |
| a, b, c (Å) | 32.573, 82.775, 34.278 | 32.514, 82.471, 34.226 | 32.515, 82.465, 34.225 | 38.942, 54.596, 128.652 |
| α, β, γ (°) | 90, 107.367, 90 | 90, 107.466, 90 | 90, 107.464, 90 | 90, 90, 90 |
| Resolution (Å) | 50.0–2.0 (2.10–2.0) | 50.0–2.5 (2.60–2.50) | 50.0–2.5 (2.60–2.50) | 50.0–2.2 (2.30–2.20) |
| No. of unique reflections | 11,332 | 5823 | 5830 | 14,417 |
| $R_{sym}$ (%) | 5.2 (14.2) | 5.3 (10.7) | 5.1 (10.3) | 11.1 (63.9) |
| $I/\sigma_I$ | 19.4 (7.8) | 16.4 (7.6) | 14.6 (7.6) | 14 (2.3) |
| Completeness (%) | 96.8 (89.1) | 97.0 (98.3) | 97.0 (98.6) | 98.3 (99.0) |
| Redundancy | 3.8 (3.4) | 2.4 (2.4) | 2.4 (2.4) | 4.9 (3.8) |
| **Refinement** | | | | |
| Resolution (Å) | 32.7–2.0 | | | 37.3–2.2 |
| $R_{work}/R_{free}$ (%) | 22.6/26.2 | | | 19.7/24.6 |
| No. of reflections | 11,303 | | | 14,179 |
| No. of atoms | 1548 | | | 1632 |
| Protein | 1480 | | | 1606 |
| Ligand/ion | — | | | 26 |
| Water | 68 | | | 90 |
| *B*-factors | 28.2 | | | 34.44 |
| Protein | 27.9 | | | 34.0 |
| Ligand/ion | — | | | 37.1 |
| Water | 32.8 | | | 42.6 |
| RMSD | | | | |
| Bond lengths (Å) | 0.007 | | | 0.008 |
| Bond angles (°) | 1.17 | | | 1.12 |
| Ramachandran | | | | |
| Favored (%) | 95.45 | | | 97.85 |
| Allowed (%) | 4.55 | | | 2.15 |
| Outlier (%) | 0 | | | 0 |

MAD diffraction data were collected from one crystal
Values in parentheses are for the highest-resolution shell

migration on native polyacrylamide gel electrophoresis (PAGE) gel in the presence of Spa2-LC (Supplementary Fig. 7d). Client valency influences protein partitioning and thus protein phase behavior[31]. We next asked how the interaction with its phase-separated binding partner Spa2 regulates the oligomer property of Aip5. We incubated full-length Aip5 with Spa2-LC in the presence of crowding agent. Surprisingly, Aip5 proteins were recruited into Spa2 liquid droplets and were therefore converted into a dynamic liquid phase from amorphous assemblies with much less motility, which was examined by fluorescence recovery after photobleaching (FRAP) experiments (Fig. 4j, k and

Supplementary Fig. 7e). At the range of physiological protein concentration and ~1:5 stoichiometry of Aip5 and Spa2 at the bud tip, Aip5 and Spa2 co-phase separated as droplet like in vitro (Supplementary Fig. 7f). We next asked how the scaffolding protein Spa2 regulates the Aip5 phase separation in vivo. Neither Aip5-C-GFP (1132–1234aa) without the N-terminus nor Aip5-N-GFP (1–1131aa) in the background of *spa2Δ* had tip or bud neck localization in vivo (Fig. 5a), consistent with the direct interactions between Aip5-N and Spa2-C (Supplementary Fig. 7d). We found the *aip5-C* (without N-terminal) showed enhanced growth defect at a stressed condition at 37 °C in the

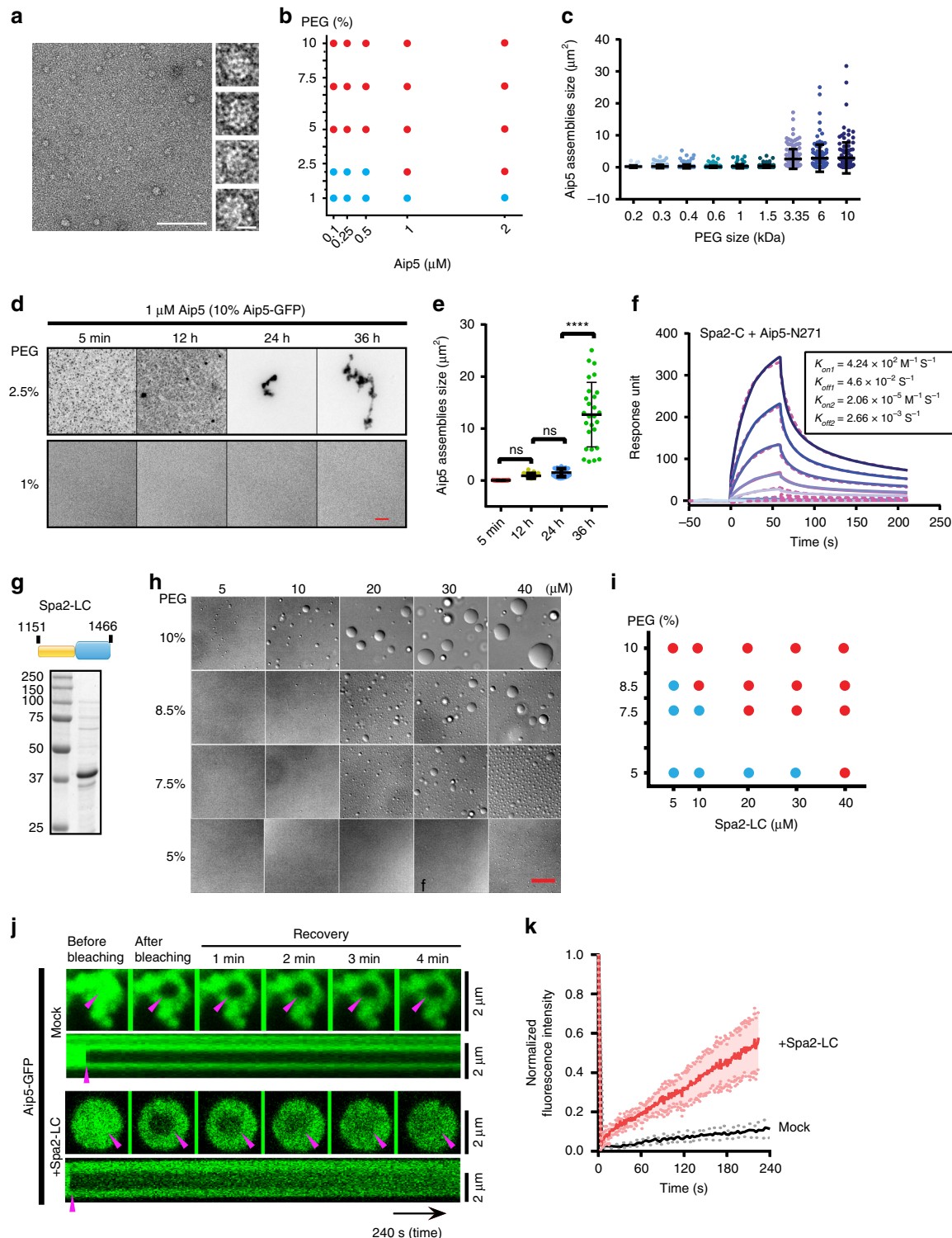

background of *bni1Δ* (Supplementary Fig. 7g), which suggested the important roles of Aip5-N in stress tolerance-related pathways.

Formation of in vivo phase-separated condensates has been implicated as an important phenomenon of IDPs during stress adaptation[35,36]. To test the phase separation of Aip5 in vivo under stress conditions, we depleted yeast of energy. Both WT Aip5 and Aip5-N proteins relocated from the bud tip and bud

neck regions into the cytosol and assembled into submicrometer-sized stress foci (Fig. 5b). Several weak Aip5-C condensates were also detected, which implies that other Aip5-binding partners, such as Bni1, might also involve in Aip5 condensation, even in the absence of Spa2. Deletion of *SPA2* caused a remarkable increase in foci number of Aip5-FL and Aip5-N but not of Aip5-C proteins (Fig. 5b, c). Both the bud tip-concentrated Aip5 at the physiological state and the induced cytoplasmic Aip5 condensate

**Fig. 4** LLPS of Spa2 regulates the phase behavior of Aip5. **a** Representative TEM images of 1 μM Aip5 in vitro under non-crowding conditions. The scale bar represents 100 nm. Insets show enlarged views of the oligomerized Aip5, and the scale bar represents 10 nm. **b** Diagram of Aip5 condensate formation, as shown in Supplementary Fig. 5e. The red dots indicate phase separation, and the blue dots indicate no phase separation of Aip5 assemblies. **c** Quantification of size from 250 nM Aip5 assemblies (10% Aip5-GFP) formed in vitro with incubation of 10% PEG (PEG200, $n = 66$; PEG300, $n = 147$; PEG400, $n = 278$; PEG600, $n = 383$; PEG1000, $n = 116$; PEG1500, $n = 274$; PEG3350, $n = 145$; PEG6000, $n = 111$; PEG10,000, $n = 119$). **d** Representative fluorescence images of 1 μM Aip5 assemblies (10% Aip5-GFP) formed in vitro by incubating with 2.5% or 1% PEG3350 at room temperature at the indicated time points. The scale bar represents 5 μm. **e** Quantification of the size of Aip5 assemblies (10% Aip5-GFP) overtime in 2.5% PEG3350, as shown in **d**. **f** The SPR sensorgram showing the binding between Spa2-C (immobilized ligand) and Aip5-N271. The correspondent curves were fitted in the bivalent model. **g** Domain schematic and SDS-PAGE gel of recombinant Spa2-LC protein. The predicted IDR and folded domain are highlighted in yellow and blue, respectively. **h** Representative DIC images of Spa2-LC-formed protein droplets under PEG3350 condition at 5 min in the protein buffer (50 mM Tris pH 8.0, 150 mM NaCl). The scale bar represents 5 μm. **i** Phase diagram of in vitro-formed Spa2-LC protein droplets shown in **h**. The red dots indicate phase separation, and the blue dots indicate no phase separation of Spa2-LC protein in DIC images. **j** FRAP assay for Aip5 assemblies, in the absence or presence of Spa2-LC in the buffer of 50 mM Tris pH 8.0 and 150 mM NaCl. ROIs were indicated by arrowhead where the kymographs are shown below. The scale bar represents 2 μm. **k** Quantification of fluorescent signals of FRAP experiments, as in **j**. ($n = 8$ for each condition.) $p$ Values were determined by the one-way ANOVA, ns not significant, ****$p < 0.0001$. Error bar, S.D. Source data are provided as a Source Data file

under energy depletion seemed to have liquid property because they could be dissolved by 1,6 hexanediol, which disrupted the weak hydrophobic interaction[37] (Supplementary Fig. 8a). However, under the same condition, Aip5 condensates formed in *spa2Δ* under energy depletion treatment could not be dissolved by 1,6 hexanediol, suggesting a less reversible nature (Supplementary Fig. 8a). Indeed, Aip5 condensates formed under energy depletion in *spa2Δ* showed the least dynamic exchange when comparing to WT under physiological or energy-depletion condition (Fig. 5d, e). Furthermore, in cells with Aip5 condensates present, we found that Spa2 co-localized with Aip5 in those condensates (Fig. 5f), where Spa2 displayed a similar dynamic property as Aip5 condensates under energy-depletion treatment (Fig. 5d, e, g, h). These results agreed with our in vitro understanding where Spa2 can improve Aip5 phase dynamic property. We additionally tested the effects of pH changes in the formation of Aip5 assemblies. Aip5 formed in vitro amorphous assemblies in a pH-dependent manner, in which a pH <5.5 induced pronounced condensed structures (Fig. 5i and Supplementary Fig. 8b). In vitro Aip5 assemblies demix with Spa2-LC liquid droplets and are dissolved over a range of pH values (pH 5.5–7.5), in which Aip5 showed lowest molecular dynamics and mobility in the phase-separated liquid droplets at the pH of 5.5 (Supplementary Fig. 8c–g). It suggests that Spa2-mediated Aip5 phase separation and motility is pH dependent. Consistently, Aip5 also demonstrated a pH-dependent phase behavior in vivo, either in the presence or absence of Spa2 (Supplementary Fig. 8h, i). In vivo FRAP assay showed that Spa2 has limited ability to reverse Aip5 condensates under the condition of pH 5.5 (Supplementary Fig. 8h, i), which agreed with the in vitro observation as well (Supplementary Fig. 8d, e). Such pH-dependent solubility of Aip5 is also consistent with the formation of Aip5 condensates under different pH conditions, where cells started to generate increasingly more obvious Aip5 condensates in the WT at pH 5.5 comparing to pH 6.5 and pH 7.5 (Fig. 5j, k). The *spa2Δ* lowered the threshold of Aip5 condensate formation from a mildly acidic condition (pH 6.5) and created a significantly higher amount of condensates than *SPA2* (Fig. 5j, k).

Interestingly, the Aip5 condensates were fully reversible in 10 min upon the addition of supplemental glucose but dissolved slower in the *spa2Δ* (Fig. 6a). Under energy-depleting condition, the *spa2Δ* also showed defects in both actin cables and cell growth (Fig. 6b–f; Supplementary Fig. 8j). The pronounced condensates of Aip5 and quick restoration upon addition of the glucose supplement motivated us to ask whether Aip5 condensation is a protective mechanism for actin assembly proteins under stress conditions. Therefore, we investigated the actin cable pattern in *spa2Δ*. Compared to WT, *spa2Δ* showed a more disruptive F-

actin under energy depletion as well as a slower recovery of the actin cables formed upon glucose supplement (Fig. 6g). *aip5Δ* showed a faster actin cable disruption (~5 min), when compared to WT (~15 min), which suggested that Aip5 involved in actin cable remodeling during stress (Supplementary Fig. 9a). However, the formation of Aip5 condensates might not be the direct factor in impairing the actin cable, because Aip5 condensates only appeared after 30 min upon energy depletion (Supplementary Fig. 9b). Furthermore, when the cells undergo prolonged stress, the Aip5 condensates without the multivalent interactions with Spa2 hampered cell growth, in which a further removal of *AIP5* in *spa2Δ* could alleviate cell growth sickness (Fig. 6f).

## Discussion
A polarisome is a multiprotein complex that is required for polarized yeast growth and the hyphal formation of filamentous fungi by regulating actin cable nucleation[38]. The IDP nature of the Aip5 and Spa2 proteins provides the dynamic assembly of macromolecular complex protein machinery in a spatial–temporal manner at the growing tip[7]. Here we report a polarisome protein Aip5 that aids in actin assembly in vitro and promotes formin Bni1-mediated actin nucleation directly through the interactions toward G-actin and the C-terminus of Bni1. The biochemical activities of Aip5 rely on the C-terminal domain, Aip5-C, which is conserved among fungi species, including various pathogenic filamentous fungi. Structural study guided our identification of an Aip5-C loop that directly interacts with G-actin, which is necessary for Aip5 actin assembly activities (Figs. 2 and 3 and Supplementary Fig. 3). The removal of the loop region abolished Aip5-C actin assembly activity as well as its promoting effect on Bni1-mediated actin nucleation, suggesting that the relative positioning of the loop region between protomers is essential in providing optimum biochemical activities. Though the actin polymerization activity of Aip5 alone in solution is relatively weak, compared to formin activity, at the physiological concentration, its promoting effect on Bni1-mediated actin assembly is robust through the binding to Bni1 C-terminal. Compared to in vitro TIRF actin assay, genetic-based growth rate using *bni1Δ aip5Δ* is less sensitive to demonstrate the synergy between Bni1 and Aip5. However, we also could not exclude possible alteration of the in vivo synergy because of the partition of other polarisome members. In vivo, polarized cell tip is crowded with macromolecular interactions of different polarisome proteins, including Bni1, Bud6, Aip5, and Spa2, which are all conserved in filamentous fungi and change protein localization and concentration during polarization, indicating potential dynamic partition coefficient for each member in such macromolecular condensates. During normal growth, the concentrated

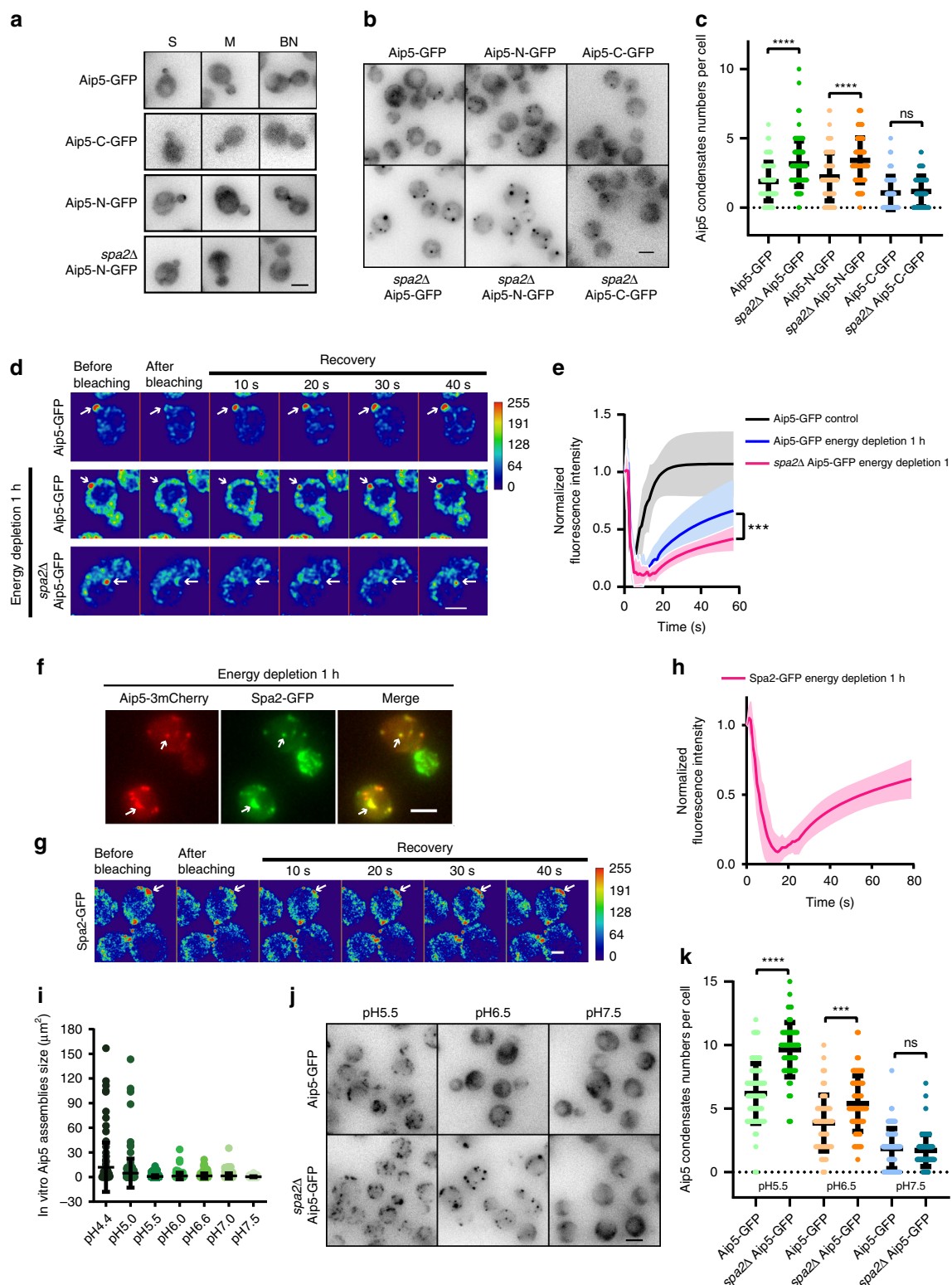

polarisome complex at the bud tip has active protein exchanges of Aip5, shown by the FRAP experiment. Scaffolder Spa2 plays particular roles in the recruitment of Aip5 to the bud tip for executing its synergistic function for actin nucleation with Bni1. At a proximate concentration and stoichiometry of Aip5 and Spa2, which are derived from fluorescent imaging and western blot approaches, Aip5 is also dissolved in Spa2 droplet in vitro. In vivo concentration measurement by more sensitive methods, such as fluorescence (cross)-correlation spectroscopy (FCCP)[39], is worth to be performed in future for a precise protein concentration measurement for each polarisome protein at the tip and in the cytoplasm. At this moment, we could not establish a definitive mechanism by which the inter-/intra-multivalent interactions of Aip5 regulates the complex nucleation activities,

**Fig. 5** In vivo condensation of Aip5 under stress conditions is Spa2 dependent. **a** Representative maximum Z-projection images of Aip5-GFP, Aip5-C-GFP, and Aip5-N-GFP protein localization in yeast. S small-budded cell, M medium size-budded cell, BN bud neck. Scale bar represents 3 μm. **b** Representative maximum Z-projection images of in vivo condensates of Aip5-GFP variants in WT and spa2Δ after 1 h of energy depletion. Scale bar represents 3 μm. **c** Quantification of the condensate number of Aip5-GFP imaged in **b**. Aip5-GFP, $n = 108$ cells; spa2Δ Aip5-GFP, $n = 126$ cells; Aip5-N-GFP, $n = 80$ cells; spa2Δ Aip5-N-GFP, $n = 154$ cells; Aip5-C-GFP, $n = 86$ cells; spa2Δ Aip5-C-GFP, $n = 82$ cells. **d** Representative deconvoluted confocal images of Aip5-GFP in FRAP assay. The scale bar represents 5 μm. **e** Quantification of fluorescent signals of FRAP experiments, as in **d**. ($n = 11$ condensates for Aip5-GFP control; $n = 11$ for Aip5-GFP under 1 h energy depletion; $n = 8$ for spa2Δ Aip5-GFP under 1 h energy depletion.) **f** Representative maximum Z-projection images of Aip5-3mCherry and Spa2-GFP localization after 1 h energy-depletion treatment; the white arrow indicates condensate formation. The scale bar represents 3 μm. **g** Representative deconvoluted confocal images of Spa2-GFP condensates (indicated by the white arrow) monitored in FRAP when cells were stressed by energy depletion for 1 h. The scale bar represents 3 μm. **h** Quantification of fluorescent signals in FRAP experiments, as in **g**. $n = 15$. **i** Quantification of in vitro formation of Aip5 assemblies (250 nM, 10% Aip5-GFP) under various pH conditions in the presence of 10% PEG3,350, as shown in Supplementary Fig. 8b. (pH4.5, $n = 95$; pH5.0, $n = 175$; pH5.5, $n = 412$; pH6.0, $n = 123$; pH6.6, $n = 216$; pH7.0, $n = 249$; pH7.5, $n = 341$). **j** Representative maximum Z-projection images of Aip5-GFP condensates formed in wild-type and spa2Δ under the indicated pH conditions. Scale bar represents 3 μm. **k** Quantification of the Aip5 protein condensates imaged in **j**. (pH5.5 Aip5-GFP: $n = 75$ cells; pH5.5 spa2Δ Aip5-GFP: $n = 55$ cells; pH6.5 Aip5-GFP: $n = 52$ cells; pH6.5 spa2Δ Aip5-GFP: $n = 70$ cells; pH7.5 Aip5-GFP: $n = 92$ cells; pH7.5 spa2Δ Aip5-GFP: $n = 81$ cells). The color key indicates signal intensity. $p$ Values were determined by the one-way ANOVA, ****$p < 0.0001$, ***$p < 0.001$, and ns not significant. All the bar graphs represent mean with an error bar, S.D. Source data are provided as a Source Data file

such as a dwell time of Aip5 and Bni1 in the Spa2 droplet[40]. In vitro reconstitution with all interacting polarisome proteins would lead to the complete mechanistic understanding. Nevertheless, here we have demonstrated that a dimeric Aip5 presents higher actin polymerization activity on its own as well as a higher NPF activity for Bni1, compared to the monomeric version of Aip5. The future work to fully reconstitute the collective behaviors of polarisome proteins would enhance our understanding of the tuneable and dynamic actin cable polymerization during the polarized cell budding and hyphal growth of yeast and filamentous fungus.

Macromolecular crowding or depletion attraction has been reported to pack actin filaments in the presence or absence of actin-crosslinking proteins[41–44] or directly accelerate actin nucleation[45]. Here, during the polarized yeast growth, Spa2 concentrates Aip5 at the growing tip where Aip5 interacts directly with G-actin and Bni1 to enhance actin nucleation. Such multivalent interactions support the in vivo maintenance of the membraneless compartment at the bud tip, by direct interaction between G-actin/Bni1 and Aip5 C-terminus, as well as Spa2 and Aip5 N-IDR. Following the polarized growth throughout the cell cycle, the concentration and localization of Aip5 are dynamically changed by switching from the tightly tip-localized condensates to the enlarging crescent on tip surface, both of which maintained a dynamic equilibrium between cytosol and the tip region. To coordinate the changes of protein localization and the maintenance of actin nucleation activities at the polarized zone during cell cycle progression, the self-assembly of Aip5 and the scaffolding function of Spa2 in recruiting Aip5 and Bni1 confer an active actin nucleation compartment by concentrating NF, NPF, and G-actin (Fig. 7). In vitro, without the crowding environment, Aip5 assemblies into functional heterogeneous oligomers in the physiological buffer. Within the solution of the crowding agent, Aip5 proteins quickly assemble into an irregular amorphous-like network. The phase behavior of Aip5 depends on protein concentration, environmental pH, and partitioning of binding partners Spa2. Under physiological conditions, client partitioning and binding by the scaffolder Spa2 protected Aip5 from solidification and solubilized Aip5 into liquid droplets, through the interactions between Spa2-C and Aip5-N, depending on the LLPS of Spa2 using the disordered residues (Fig. 7). Similar molecular crowding-regulated scaffolder–client interaction in LLPS was also found for microtubule (MT) assembly in *Caenorhabditis elegans*. A pericentriolar material scaffold protein (SPD-5) forms condensates and recruits the client MT effector proteins for nucleating MT[46,47].

Global coordination of intracellular processes is complicated by various physiological stresses, including thermal, pH, and energy-depletion conditions, which could lead to acute protein folding change and aggregate formation. Stress-induced irreversible protein aggregation in vivo is thought to drive a liquid–solid protein phase transition and therefore inactivates protein function, which is destructive, whereas the formation of reversible protein aggregates was proposed to serve as a protective mechanism, such as preventing protein degradation and aiding in efficient recovery for immediate use after restoration of favorable conditions[35,36,48]. IDPs are crucial components for receiving extracellular and intracellular signals to fine-tune the protein–protein interactions and biochemical activities[49,50]. Spa2 constrains the formation of Aip5 condensates under energy depletion and facilitates the reversal of Aip5 condensates and the repolarization to budding tip with energy restoration (Fig. 7). Currently, our results suggest coordinated functions and behaviors of Aip5 under physiological and stress conditions. During the harsh energy-depleting circumstances, cells become quiescent for short-term survivability. At the same time, Aip5 starts to form in vivo condensate, through increasing intramolecular and intermolecular interactions, which might decrease Aip5 proteostasis as a rapid adaptive response under stress conditions, which temporarily attenuated Aip5 biochemical activities in actin assembly. Therefore, before Aip5 assembly into large in vivo condensates, actin cable structures start to be compromised quickly (Supplementary Fig. 9a, b). Once favorable growth conditions are encountered, Spa2-mediated dissolution of Aip5 condensates confer a quick reversal of Aip5 functions in actin assembly. Different from the less mobile phase of recombinant Aip5 on its own in vitro, the in vivo Aip5 condensates that were formed by energy depletion are soluble by energy restoration, which is still fine-tuned by Spa2. It implies a sophisticated and adaptive proteostasis network in vivo that modulate Aip5 condensation, which is reversible by glucose supplement. In such orchestrated multiprotein complex, Spa2 is a primary determinative factor that controls the phase and size of Aip5 condensates. More other binding partners are also involved in regulating Aip5 condensation in vivo since Aip5 condensates are still able to be dissolved in spa2Δ under stress condition. Polarisome protein Bni1 and Bud6 are also likely involved in assembling Aip5 condensates since both of them tuned in vivo Aip5 localization (Fig. 1a, b). However, when undergoing a prolonged period of adverse condition, in vivo Aip5 condensates become detrimental that influence cell viability gradually. Currently, the mechanism by which the bi-phase function of the phase-separated Aip5, or

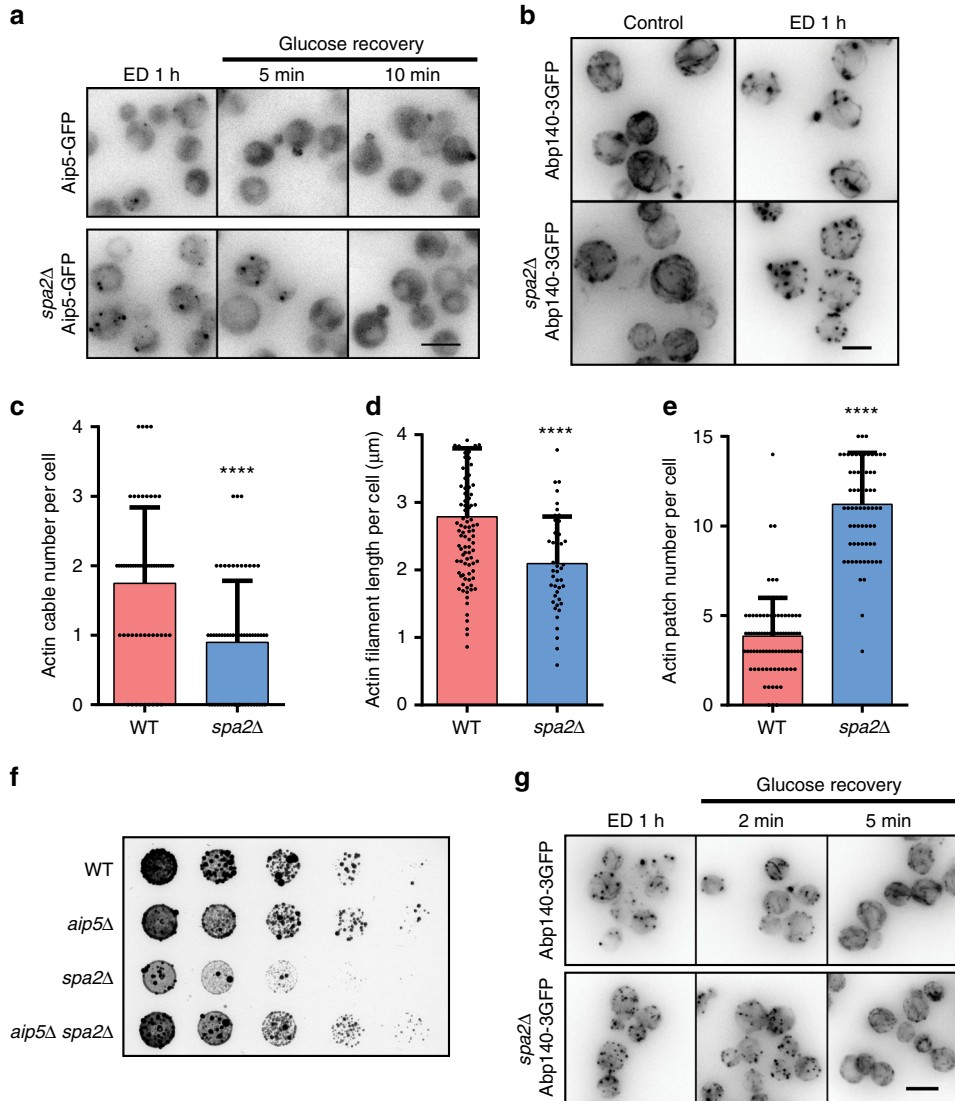

**Fig. 6** Reversible in vivo condensation of Aip5 under stress condition. **a** Representative maximum Z-projection images of in vivo formed Aip5-GFP condensates under 1 h of energy-depletion treatment and followed by replenishment of 2% glucose. The scale bar represents 5 μm. **b** Representative maximum Z-projection fluorescence images of Abp140-3GFP revealed actin cable pattern in WT and spa2Δ, with or without 1 h of energy depletion. Scale bar represents 3 μm. **c–e** Quantification of the Abp140-3GFP marked actin cables and actin patches that were imaged in **b** under energy-depletion treatment for 1 h (cells for analyzing actin cable number: n = 59 for WT and spa2Δ; Actin cable length: n = 99 for WT and n = 45 for spa2Δ; cells for analyzing actin patch number: n = 87 for WT and n = 70 for spa2Δ). **f** Yeast growth assay on YPD plate with 3.1 mM 2-deoxy-D-glucose and 0.31 μM antimycin A for 72 h at 30 °C. **g** Representative maximum Z-projection images of Abp140-3xGFP-labeled actin cables from the indicated strains after 1 h of energy depletion and recovery with replenishment of 2% glucose. Scale bar represents 5 μm. WT wild type. p Values were determined by two-tailed Student's t test assuming equal variances, ****p < 0.0001. All the bar graphs represent mean with an error bar, S.D. Source data are provided as a Source Data file

other protein condensates, are balanced at the early and late stages of cell stress is not clear. The intrinsic molecular grammar, constituent participation, and impaired protein proteostasis, which is a hallmark of age-associated proteinopathies[51], might be the underlying factors. Future in vitro reconstitution of the macromolecular complex with polarisome members and identification of potential molecular chaperones is likely able to examine these possible mechanisms.

## Methods

**Yeast strain construction and yeast cell culture**. *Saccharomyces cerevisiae* S288C strains with endogenous C-terminal tagging were created essentially by lithium acetate transformation[39,52]. First, YMY2011 was transformed with dsDNA that encoded a fluorescent fusion protein (XFP) gene and with 50 bp flanking sequences upstream and downstream of the targeted locus. The transformants were selected

on either YPD agar plates containing 200 μg/ml Geneticin or synthetic complete (SC) and drop-out medium based on the selectable marker. Deletion of the specific gene was achieved by replacing the gene open reading frames with the *Candida glabrata LEU2*, *HIS3*, *URA3*, or *KanMX* cassettes. Haploid strains were obtained by sporulation and tetrad dissection and selected by various selectable marker genes. The yeast stain expressing Aip5 C-terminal domain was generated using integration plasmid pRS305 that encodes the *YFR016C* DNA fragment 3394–3702 bp and was selected against the SC drop-out leucine plates. All the yeast strains generated from this study are listed in Supplementary Table 1. Generally, yeast cells were grown in YPD (10 g per liter yeast extract, 20 g per liter peptone, 20 g per liter glucose) medium or 2% agar plates at 30 °C or using synthetic media (SM) supplemented with appropriate amino acids[20,52,53].

**Cell growth assay**. For spotting assay on the YPD plate, each tested yeast strain was firstly streaked out from -80°C glycerol stock. Yeast was cultured in YPD liquid medium overnight. Yeast was reinoculated and cultured in YPD medium the next morning for 6h from optical density (OD600) = 0.2. Cultures were adjusted to the

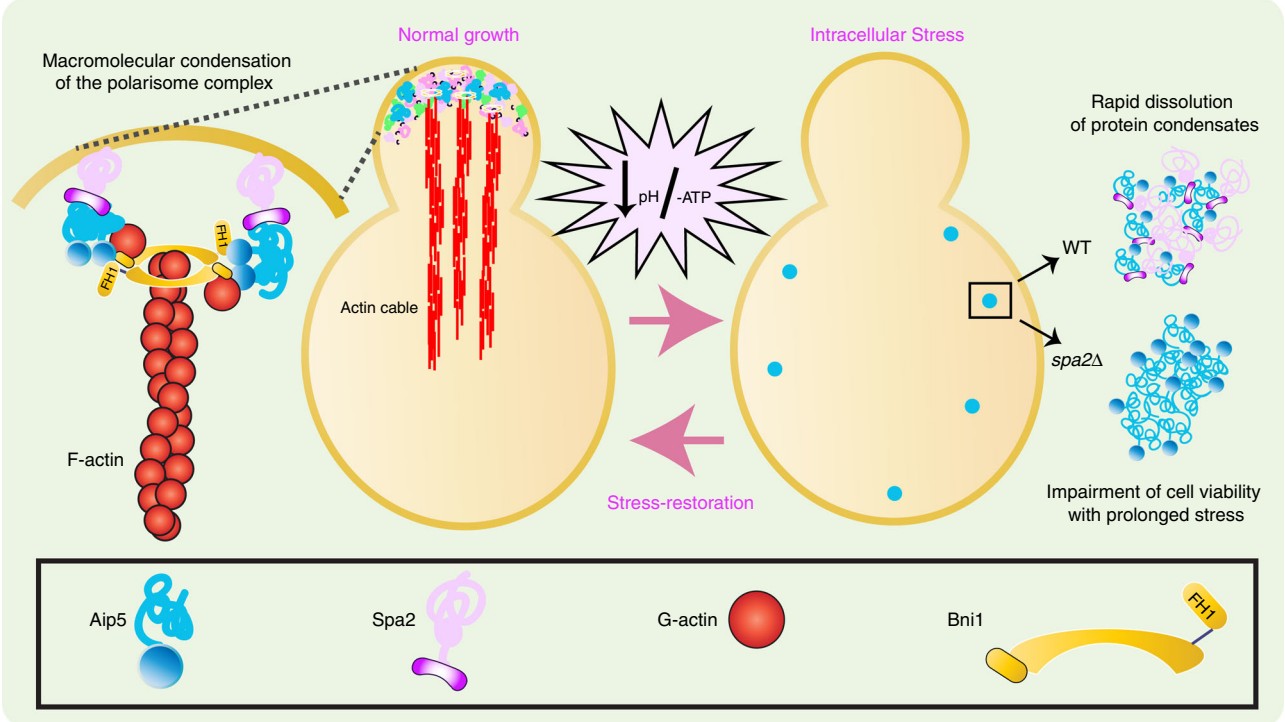

**Fig. 7** Cellular behavior of a polarisome complex protein Aip5. Under physiological growth, Spa2 recruits Aip5 to non-membranous polarisome complex at the bud tip for actin cable assembly together with Formin Bni1. Under stress conditions, Aip5 concentrates into in vivo reversible condensates, which undergo reversible LLPS in the cytoplasm in a Spa2-dependent manner. The Aip5 condensates showed compromised biochemical activity in polymerizing actin during the stress response and but reversed for actin assembly upon stress restoration

same initial starting OD600 = 1.0 (corresponds to 8–9 × 10$^{-6}$ cells/ml). Tenfold serial dilution was prepared with fresh YPD, and 4 µl aliquots of undiluted or diluted cells were spotted on the YPD agar plate. Plates were incubated at the tested temperatures for 36 h and scanned by an Epson v600 scanner at 300dpi. Grayscale images of each plate were generated by Photoshop. For energy-depletion assay, YPD agar plates were supplemented with 3.1 mM 2-deoxyglucose (2-DG; Sigma) and 0.31 µM antimycin (Sigma) and grown at 30 °C for 72 h before scanning.

For cell growth liquid assay with LatA (I-DNA Biotechnology), a dense overnight culture of each strain was reinoculated into fresh YPD and allowed to grow for 4 h. The liquid yeast cultures were adjusted to OD600 = 0.2 in 130 µl in a 96-well transparent bottom microplate sealed with a Breathe-Easy® sealing film (Diversified Biotech). Each tested condition contained 4 replicates that were growing at 25 °C for 22 h. The OD600 of each well was measured and recorded by the TECAN m200pro plate reader machine every 15 min, and the plate was kept shaking between each measurement to avoid cell precipitation. The collected data were subjected to GraphPad Prism 6 (GraphPad, San Diego, CA, USA) to generate a growth curve with an averaged OD600 and error bars in S.D.

**Protein expression and purification.** Aip5 proteins containing the IDR region were purified from the yeast system[34,52]. In brief, the overexpression protein constructs were transformed to YMY2043 and selected by synthetic minimal (SM) drop-out Ura3 plate (2% glucose). And the positive colonies were cultured from small scale (5 ml) till large scale (1 liter) in SM drop-out Leu2 medium with 2% raffinose until saturation at 30 °C. To induce the protein expression, 10× YP (100 g per liter yeast extract, 200 g per liter peptone) was added to a final of 1× and 2% galactose was supplemented. The yeast cultures were cultivated at 30 °C for overnight and collected by centrifugation from JA-10 rotor (Beckman) at 6300 × g for 10 min. The collected cells were washed one time by sterilized H$_2$O and freshly frozen by liquid N$_2$ and then lysed by cryomilling machine (SPEX SamplePrep 6870 Freezer/Mill). The lysed cell powder can be stored in −80 °C freezer before protein purification. Five grams of lysed cell powder was used each time for protein purification, where they were dissolved in 50 ml buffer A (50 mM Tris pH 7.6, 500 mM NaCl, 20 mM Imidazole) with 5 µl protease inhibitor cocktail (BioWorld). The lysates were clarified by centrifugation at 40,000 × g for 1 h by rotor JA25.5 (Beckman). And the supernatant was filtered by 0.22-µm membrane before applying to the Ni-NTA affinity chromatography. Fractions of target proteins were eluted by imidazole gradient from buffer B (50 mM Tris pH 7.6, 500 mM NaCl, 500 mM Imidazole) and concentrated to <5 ml. Furthermore, the proteins were purified in 50 mM Tris pH 7.6, 500 mM NaCl by size-exclusion chromatography

using a HiLoad 16/600 Superdex200 column (GE Healthcare). The target protein fractions were collected and concentrated before freshly frozen by liquid N$_2$.

Bni1FH1COOH, Bni1FH2C, Bni1FH1FH2, Bni1FH2, Bni1-C, Cof1, Aip5-mC, and Spa2-LC were expressed in *Escherichia coli* (BL21(DE3) Rosetta T1R) and purified from 2 l of TB medium (24 g per liter yeast extract, 20 g per liter tryptone, 4 ml per liter glycerol, phosphate buffer pH 7.4). To express Aip5-C, the expression plasmids containing Aip5-C variants were transformed into *E. coli* strain BL21 (DE3) through heat shock. Bacterial cells carrying the recombinant plasmids were grown in 2×YT medium (10 g per liter yeast extract, 16 g per liter tryptone, 5 g per liter NaCl, final pH 6.8 ± 0.2) with the addition of 50 µg/ml of Kanamycin at 37 °C to an OD600 of 0.6. Protein expression was induced by 0.5 mM IPTG at 16 °C overnight. The same *E. coli* strain was used for transformation of Selenomethionine-labeled Aip5-C. Cells were grown at 37 °C in M9 minimal medium complemented with glucose, standard trace elements, and the corresponding antibiotic. The temperature was cooled down to 16 °C when the OD600 of the cells reached 1.0. A mixture of amino acids except L-methionine was added, followed by the addition of 0.025 mg/ml of L-selenomethionine. The cells were allowed to grow for 20 min before induction with 0.5 mM IPTG at 16 °C overnight. The cells were harvested by centrifugation at 4 °C, 5000 × g (rotor JA10) for 15 min. The pellet was resuspended and lysed by EmulsiFlex-C3 (www.avestin. com). The lysate was clarified by centrifugation at 40,000 × g for 40 min by rotor JA25.5. The supernatant was purified by Ni-NTA affinity chromatography. Fractions of the target protein were pooled following gradient elution with increasing imidazole concentration. The protein was dialyzed against a buffer containing 20 mM Tris-HCl pH 8.0, 150 mM NaCl, with the addition of TEV protease to cleave the His6-tag. A second Ni-NTA affinity chromatography was performed to remove the cleaved His6-tag and TEV protease. The protein was further purified by size-exclusion chromatography using a HiLoad 16/600 Superdex75 column (GE Healthcare) in 20 mM Tris-HCl pH 8.0 and 150 mM NaCl. The target protein was concentrated to a final concentration of 18.5 mg/ml for crystallization screens.

To purify monomeric Ca$^{2+}$-ATP-actin and label with Oregon Green™ 488 Iodoacetamide (ThermoFisher) or NHS-dPEG®4-biotin (Sigma), 5 g of rabbit muscle acetone powder (Pel-Freez Biologicals) was used to dissolve in 60 ml of G-buffer (5 mM Tris pH 8.0, 0.2 mM ATP, 0.1 mM CaCl$_2$, 0.5 mM dithiothreitol (DTT)) at 4 °C for 30 min, then filtered the solution through cheese cloth to collect the actin-rich extracts. The filtered rabbit muscle powder was repeated with the previous procedure for another three times to collect actin-rich extracts. The combined actin-rich extracts were centrifuged at 18,000 × g by rotor JA-25.5 (Beckman) to collect the supernatant. Afterwards, the solution was stirred slowly at 4 °C and 50 mM KCl and 2 mM MgCl$_2$ was added to allow actin polymerization for

1 h. To remove tropomyosin and other protein from F-actin, KCl powder was used to add slowly into the F-actin solution with a final concentration of 0.8 M and the solution was stirred slowly for another 30 min. To collect the polymerized F-actin, all the solution was subjected for centrifugation at 4 °C, 95,800 × g with rotor Ti45 (Beckman) for 3 h. F-actin pellet was rinsed with 1 ml G-buffer and all the F-actin pellets were transferred to the 10 ml homogenizer with 5 ml G-buffer and the pellet was homogenized by moving up and down. To further depolymerize the F-actin, we applied sonication with 3 s on, 10 s off and repeated the cycle for 4 times. Afterwards, we dialyzed the actin against 1 liter of G-buffer without DTT for overnight. The next day, we changed to 1 liter of new G-buffer without DTT and kept on dialysis. In the meanwhile, we freshly prepared the Oregon Green™ 488 Iodoacetamide or NHS-dPEG®4-biotin with high-quality dimethylformamide into a final concentration of 10 mM. The dialyzed actin was taken out and the concentration was determined through nanodrop at reading OD290. Before proceeding to actin labeling, we diluted the actin with cold 2× labeling buffer (50 mM Imidazole pH 7.5, 200 mM KCl, 0.3 mM ATP, 4 mM MgCl$_2$) until the concentration was 23 µM. Afterwards, we added 12–15-fold molar excess of Oregon Green™ 488 Iodoacetamide/NHS-dPEG®4-biotin stock dropwise while very gently vortexed. To allow sufficient labeling, we kept the solution in the dark with aluminum foil and rotated overnight at 4 °C. The next day morning, we pelleted the labeled actin with Type 50.2 rotor (Beckman) at 111,000 × g for 3 h. We then collected all the pellets and transferred to the homogenizer with 5 ml G-buffer. Afterwards, the actin was homogenized for 20 times and sonicated with 4 cycles of 3 s on, 10 s off. To further depolymerize the actin, we dialyzed against 1 liter of G-buffer for at least 2 times (24–36 h). We then applied the dialyzed actin to centrifugation at 167,000 × g for 2.5 h with rotor SW55 Ti (Beckman) at 4 °C. To further purify the actin, we collected the 2/3 of the supernatant on top and injected to fast protein liquid chromatography system, separated through column HiPrep™ 16/60 Sephacryl™ S-300 HR. The collected labeled actin was dialyzed overnight against G-buffer with 50% glycerol to reduce the total volume. Small aliquots of actin were prepared and freshly frozen by liquid N$_2$ for long-term storage.

**Pyrene-actin assembly.** Pyrene-labeled actin was purchased from Cytoskeleton Inc. A 10 µM G-actin (5% pyrene actin) was first converted to Mg$^{2+}$-ATP-actin for 5 min on ice before use and then mixed rapidly with various proteins in the G buffer. The spontaneous actin polymerization was initiated by 10 × KME buffer mix (10 mM MgCl$_2$, 10 mM EGTA, and 500 mM KCl), at a total reaction volume of 120 µl. The pyrene-actin fluorescence signal was monitored in a plate reader Cytation 5 (BioTek, USA) at excitation and emission wavelengths of 365 and 407 nm, respectively. Initial relative polymerization rate in the pyrene assay was quantitatively analyzed[22,54,55]. In brief, the raw data from 60 to 180 s were used to generate a straight line that is fitted with a slope with the $R^2$ value >0.95. The fold change of actin polymerization rate was normalized by dividing each slope of the tested condition using the slope of the control curve. The values shown in the graph were the average data from at least two times of independent experiments.

To test the capping releasing effect, 0.4 µM capped actin seeds were mixed with 0.4 µM G-actin (5% pyrene actin) together with the desired proteins. All the reactions were initiated by adding 10× KME buffer.

For actin depolymerization assay, 5 µM of actin (30% pyrene actin) was polymerized in the F-buffer at room temperature for 2 h before diluted to 0.1 µM. The decrease of pyrene fluorescence was monitored in the presence of Cof1 or Aip5. The actin polymerization graphs were plotted by the GraphPad Prism 6.

**Total internal reflection fluorescence microscopy.** For TIRFM experiments, 25 × 50-mm coverslips (Marienfeld Superior) were cleaned by 20% sulfuric acid for overnight and rinsed thoroughly by sterile water. Afterwards, the coverslips were coated with 2 mg/ml methoxy-PEG-silane and 2 µg/ml biotin-PEG-silane (Laysan Bio Inc.) in 80% ethanol (pH 2.0, adjusted by HCl) at 70 °C for overnight. The next day, coverslips were rinsed thoroughly by sterile water and dried in N$_2$ stream, which can be kept at −80 °C for long-term storage. Before each experiment, the functionalized coverslip was attached to a plastic flow cell chamber (Ibidi, sticky-Slide VI 0.4). Afterwards, the flow cell was incubated for 30 s with HBSA (20 mM Hepes, pH 7.5, 1 mM EDTA, 50 mM KCl, and 1% bovine serum albumin), incubated for 60 s in 0.1 mg/ml streptavidin in HEKG10 (20 mM Hepes, pH 7.5, 1 mM EDTA, 50 mM KCl, 10% [vol/vol] glycerol). Finally, the flow cell chamber was washed by 1× TIRF buffer (10 mM imidazole, 50 mM KCl, 1 mM MgCl$_2$, 1 mM EGTA, 0.3 mM ATP, 50 mM DTT, 15 mM glucose, 100 µg/ml glucose oxidase, and 0.5% methylcellulose [4000 cP], pH 7.4). Proteins in TIRF buffer were mixed with 0.5 µM G-actin (10% Oregon Green 488 labeled, 0.5% biotin labeled) and then flowed into the chamber. Images were acquired at room temperature at 5-s intervals for 10 min using Apochromat TIRF 100XH NA 1.49 (Nikon Instruments) on Nikon ECLIPSE Ti-S inverted microscope with iLAS2 motorized TIRF illuminator (Roper Scientific, Evry Cedex, France). The illumination source, microscope stage, and an Evolve 512 EMCCD camera (Photometrics, Tucson, AZ) are all under the control of MetaMorph 7.8 software (Molecular Devices, Sunnyvale, CA). The focus was maintained using the Perfect Focus System. To quantify the actin filament elongation rate, the individual filament in each sample was traced by hand for at least 2 min each, and the measured actin filament length was then divided by the correspondent time to determine the elongation rate. We used the conversion factor of 370 subunits per micrometer of F-actin to calculate the barbed-end elongation rate.

**Live-cell fluorescence imaging.** Yeast strains were cultured overnight at 25 °C in the synthetic complete media without tryptophan (to minimize auto-fluorescence) and reinoculated into fresh medium to a final OD600 = 0.15–0.2. Cells were allowed to grow for an additional ~4 h before imaging. Cells were immobilized onto Concanavalin A (1 mg/ml)-coated coverslips and imaged by wide-field microscope Leica DMi8 (Leica Microsystems, equipped with ORCA-Flash4.0 LT scientific CMOS camera (Hamamatsu Photonics, Japan) and Leica ×100 oil immersion objective lens (NA 1.4). For whole-cell imaging, images were acquired continuously at a 0.25-µm interval for a total range of 7.5 µm in the z-direction, using an exposure time of 200 ms and 1× binning. To quantify the lifetime of Abp1-mRFP-labeled actin patches, images were acquired continuously at rates of 1 s per frame and 90 frames in total. To examine in vivo Aip5 aggregation in response to energy depletion or pH changes, early log-phase cells in SM drop-out tryptophan were treated under the conditions of 20 mM 2-DG (inhibition of glycolysis) and 10 µM antimycin A (inhibition of mitochondrial ATP production) or 100 mM phosphate-buffered saline (PBS) pH 5.5–7.5 containing 2 mM 2,4-dinitrophenol (Sigma). Cells were incubated for 1 h before imaging. The glucose recovery assay was conducted by removing the original culture medium and supplemented with 2% glucose in the synthetic complete media.

**In vitro actin polymerization imaging.** A 10 µM G-actin prepared in G-buffer was converted to Mg$^{2+}$-ATP-actin on ice for 5 min first and then mixed rapidly with proteins in the G buffer. The actin polymerization was initiated by 10× KME buffer mix in a total reaction volume of 50 µl for 30 min. Five µl polymerized actin sample was incubated with acti-stain™ 488 phalloidin (Cytoskeleton, Inc.) at a final concentration of 0.5 µM for 5 min before being diluted by F-Buffer (G-buffer plus 1× KME) and applied on the poly-lysine (0.01%)-coated cover glass for imaging using a ×63 oil objective lens. To analyze the fluorescent phalloidin-labeled actin filament in vitro, the individual filament was manually tracked and auto-aligned by the software auto-detection using ImageJ plugin 2D Jfilament.

**In vitro protein condensate assembly and imaging.** Aip5 proteins (10% Aip5-GFP) were incubated with 10% PEG in the physiological buffer (50 mM Tris pH 7.6, 150 mM NaCl) for 5 min at room temperature, and 5 µl of the reaction were applied on the cover slides and fluorescence imaging was conducted by a wide-field microscope using ×100 oil immersion objectives. The different molecular weight of crowder agent PEG was prepared in the protein storage buffer (50 mM Tris, pH 7.6, 150 mM NaCl) at a stock concentration of 50% (w/v). For testing various pH effects on the formation of protein condensates, buffers of pH 5.0–7.5 were prepared in 50 mM PBS and buffer pH 4.4 was prepared in 50 mM sodium acetate. Two µM of Aip5 proteins (10% Aip5-GFP) were mixed rapidly in the different pH buffers in the presence of 10% PEG3350 and incubated at room temperature for 5 min before imaging. For capturing the differential interference contrast images of the Aip5 truncating protein variants, 2 µM of purified protein was incubated with 10% PEG3350 in the 50 mM Tris, pH 7.6, and 150 mM NaCl for 5 min at room temperature before imaging. To quantify the various protein condensate size formed in the phase transition manner, fluorescent Aip5 condensates under different crowding conditions were selected by ImageJ using threshold function and subjected to size measurement by the particle analysis plugin. The phase diagram was generated by plotting the size of Aip5 assemblies over varied conditions using GraphPad Prism 6.

**Fluorescence recovery after photobleaching.** FRAP experiments were performed on an Carl Zeiss LSM 710 with FRAP wizard, using a ×100 oil objective lens. Condensates of 500 nM Aip5 (10% Aip5-GFP) was mixed with 30 µM Spa2-LC in the condensate buffer (50 mM Tris, pH 7.6, 150 mM NaCl) for 5 min before adding the PEG3350 to a final concentration of 10%. In vivo Aip5 condensates were introduced by stressed cells under energy depletion or various pH treatments. Protein assemblies or liquid droplets were monitored by time-lapse imaging with 500-ms exposure and 1-s interval for 250 cycles, and the in vivo Aip5 condensates were monitored with 500-ms exposure and 1-s interval for 80 cycles. The 100% bleaching by 488-nm laser was achieved by three cycles of scanning, and the bleaching was stopped once the fluorescence signal dropped to 40% comparing to original signal intensity. For FRAP analysis, we measured mean fluorescence intensities from three regions of interests (ROIs), including one in the photobleached region, one outside of droplet for correcting illumination-caused photobleaching, and one in the background region. Fluorescence recovery was analyzed using the plugin FRAP Profiler of ImageJ and plotted by GraphPad Prism 6.

**Negative stain and TEM.** Negative staining of various purified Aip5 proteins was performed by a direct application of 4 µl of proteins, at a concentration of 1 µM, to the glow discharged carbon-coated copper grid (200 mess, Electron Microscopy Sciences Inc. USA), followed by 30 s to 3 min of uranyl acetate (0.5–2% in water) staining. Air-dried samples were examined at 120 kV in a Tecnai 12 TEM, and images were recorded using an Ultrascan 1000 CCD camera (Gatan, Inc.). For

Supplementary Fig. 5c, 1 μM Aip5 was mixed with or without 500 nM arginine and incubated on ice for 15 min before applying to the grid.

**Crystallization**. Crystal screening for native Aip5-C, SeMet Aip5-C, and Aip5-C-3m were carried out through sitting drop vapor diffusion method using 96-well intelli-plate® (Art Robbins Instruments). Gryphon LCP (Art Robbins Instruments) and Nanoliter Pipetting System mosquito® (TTP LabTech) were used to perform the screening. Three drops of different protein-to-reservoir ratios (0.2 μl:0.1 μl, 0.15 μl:0.15 μl, 0.1 μl:0.2 μl) were set up per condition in each well of the intelli-plate®. The plates were incubated in RockImager (Formulatrix) at 20 °C for over 2 weeks. Native Aip5-C crystal was crystallized in 0.1 M Tris-HCL pH 7.0, 20% PEG1000 with protein-to-reservoir ratio at 2.0 μl to 1.0 μl. SeMet Aip5-C crystal was crystallized in 0.2 M potassium sodium tartrate and 20% PEG3350 with a protein-to-reservoir ratio at 2.0 μl to 1.0 μl. Aip5-C-3m was crystallized in sodium phosphate citrate pH 4.2 and 40% PEG300 with a protein-to-reservoir ratio at 2.0 μl to 1.0 μl. Crystals were picked directly from the drop and cryoprotected in reservoir solution supplemented with 15–25% v/v glycerol except for Aip5-C-3m, which did not require cryoprotectant. Crystals were flash-frozen in liquid nitrogen before data collection.

Diffraction data for native Aip5-C and MAD data for SeMet Aip5-C were collected at BL13B1 beamline of the National Synchrotron Radiation Research Centre, Taiwan. The dataset for Aip5-C-3m was collected at an in-house X-ray facility using Rigaku FR-X. The position of selenium site was located by SHELDX[56], and the initial experimental phase was calculated using PHENIX AUTOSOL, followed by model building in PHENIX AUTOBUILD. The collected diffraction data of Aip5-C and Aip5-C-3m were indexed, integrated, and scaled using HKL3000. Structure of Aip5-C-3m was determined by molecular replacement with Phenix_PHASER using the native Aip5-C structure as a search model. All structures were built using PHENIX AUTOBUILD and manually improved in COOT. Subsequent refinement was performed with a combination of COOT and PHENIX.REFINE. Water molecules were included in the last round of refinement. The crystallographic statistics of the structures are summarized in Table 1. All structural figures were prepared with PyMol (https://www.schrodinger.com/pymol, Schrödinger, LLC). The crystal structures of native Aip5-C and mutant Aip5-C-3m are deposited in the PDB under the accession codes 6ABR and 6ABS, respectively.

**Characterized Aip5 variants by size-exclusion chromatography**. To quantify the Aip5-C solution molecular weight, 100 μl of 2 mg/ml purified Aip5-C were ran two times on the calibrated column superdex 75 10/300 GL (GE Healthcare) in buffer containing 20 mM Tris pH 8.0 and 150 mM NaCl. The first time Aip5-C protein was injected alone into the column; the second time Aip5 protein was mixed with 4 standard proteins (Blue dextran: 2000 kDa; Conalbumin: 76 kDa; Ovalbumin: 43 kDa; and Ribonuclease A: 13.7 kDa) and injected into the column. The standard curve of protein size (yAxis: LogDa) was plotted against the standard protein elution volume (xAxis: ml). As a result, Aip5-C protein size in the solution can be calculated based on its elution volume (around 12 ml). The UV280 elution profiles were plotted in GraphPad Prism 6.

To test the arginine effects in perturbing Aip5 oligomerization, 500 μl of 1 μM Aip5 was incubated with or without 500 nM arginine on ice for 15 min before subjecting to the size-exclusion chromatography using calibration column superdex 200 increase 10/300 GL (GE healthcare). The Aip5 elution profiles from two conditions were plotted and merged for comparison volume in GraphPad Prism 6.

**Yeast cell protein extraction and immunoblotting**. Yeast whole-cell protein extracts were prepared by trichloroacetic acid (TCA) precipitation method[57]. In brief, freshly cultured yeast cells from total OD600 = 10 were collected through centrifugation at 3000 × g for 5 min and resuspended to 250 μl 20% TCA with 100 μl 0.5-mm glass beads (Biospec). The cells were lysed by powerlyzer (Qiagen) at 3500 rpm for 30 s in 3 cycles with 1-min interval each on ice. And the cell lysates were transferred out and the glass beads were washed by 300 μl 5% TCA to collect remaining cell lysate. Furthermore, 700 μl 5% TCA were added to the cell lysates before centrifuging at 14,000 rpm by Eppendorf™ Benchtop Microcentrifuge (Thermo Fisher). The pelleted proteins were resuspended to 40 μl 1 M Tris pH 8.0 and 80 μl 2× sodium dodecyl sulfate (SDS) loading dye before boiling at 95 °C for 10 min. Then the protein samples were loaded to SDS-PAGE gel and detected using western blot with the following primary antibodies: mouse anti-Pgk1 (1:10,000; Invitrogen, Cat. # 459250), mouse anti-c-Myc (1:1000; 9E10, Cat. # 13–2500), and rabbit anti-GFP (1:500; Torrey Pines Biolabs, Inc., Cat. # TP401). The membranes were probed with secondary antibody IRDye® 800CW Goat anti-Mouse (1:15,000; LI-COR, Cat. # 925–32210) or IRDye® 680RD Goat anti-Rabbit (1:15,000; LI-COR, Cat. # 925–32211). Blots were subsequently scanned on an Odyssey Infrared Imager (LI-COR Biosciences, USA) (also see Supplementary Fig. 10).

**Microscale thermophoresis**. Binding affinity between actin and Aip5 loop peptide (loop: TSLAGGGFHM) was measured by the MST method. Aip5 peptide was mixed with 2 μM G-actin at a 1:1 ratio in a total volume of 20 μl, where various stock concentrations of Aip5 peptides were prepared with the highest

concentration as 2 mM. The binding reactions were carried out in G-buffer with an additional 0.05% Tween 20 at room temperature. Samples were loaded into Monolith™ NT.115 standard capillaries (Nanotemper Technologies), which were incubated at 25 °C in the Monolith™ NT.115 apparatus (Nanotemper Technologies) before measurement. The data were collected at 25 °C using the red LED at 5% (GREEN filter; excitation 515–525, emission 560–585 nm) and IR-Laser power at 80%. Data analyses and the binding curves were fit with Kd mode in the MO analysis software (Nanotemper Technologies).

**Fluorescence anisotropy**. To determine Aip5-C and Aip5-C-LD affinity toward candidate binding partners, purified Aip5-C or Aip5-C-LD protein was labeled by amine-reactive Alexa Fluor 488 dye (ThermoFisher scientific). The total reaction volume was 25 μl containing 30 nM Alexa Fluor 488-labeled Aip5-C or Aip5-C-LD. The ligands to be titrated into Aip5-C or Aip5-C-LD were prepared in different stock concentrations and added as 1/5th of the final reaction volume (5 μl). The non-polymerizable monomeric actin (Hypermol, Germany) was titrated down starting from 50 μM, whereas Bni1 protein variants were started from 5 μM. Reactions were incubated in low-volume non-binding black 384-well plates at room temperature for 2 h. FA was measured by the plate reader Synergy™ H4 (BioTek, USA) using fluorescence polarization mode (excitation filter: 485 nm, 20 nm bandpass; emission filter: 510 nm, 20 nm bandpass). The binding curves were fitted using the Hill equation in GraphPad Prism 6 with the means from at least three independent experiments.

**Circular dichroism**. Eighty μl of 0.5 mg/ml purified Aip5 proteins were prepared in a buffer (20 mM Tris pH 7.6, 50 mM NaCl) and then loaded in the water-jacketed, 1 mm path length cylindrical quartz cuvette (Hellma). The CD spectra were collected using a Jasco J-710 spectropolarimeter at 0.5 nm resolution and a scan rate of 200 nm/min at room temperature. Reported spectra were averages of three scans and smoothed. Reported molar ellipticities were calculated by subtracting the background spectrum in the Chriascan CD Spectrometer.

**Surface plasmon resonance**. SPR experiment was performed by Biacore T200 instrument (GE Healthcare) at room temperature in buffer containing 20 mM HEPES and 150 mM NaCl pH 7.4. Ligand protein Spa2-C was immobilized on the CM5 chip (GE Healthcare) by amine coupling. The carboxyl group on the dextran surface of the chip was converted to amine-reactive ester by reacting with 0.2 M 1-ethyl-3-(3-dimethylpropyl)-carbodiimide and 0.1 M N-hydroxysuccinimide. The ligand to be immobilized was then injected to the surface at a flow rate of 10 ml/min at pH 4.5, while the reference cell was left blank without the injected protein. To test the binding of analyte Aip5-N271 with the immobilized ligand Spa2-C, serially diluted (1:1) Aip5-N271 was flown in over the surface of the control and ligand for 60 s and dissociated with buffer (20 mM HEPES, 150 mM NaCl, pH 7.4) for 150 s at a rate of 30 μl/min. Aip5-N271 was injected at gradient concentrations as 30, 15, 7.5, 3.75, 1.875, 0.934, 0.469, and 0.234 μM. The chip surface with left-over protein captured was regenerated by treating with 50 mM NaOH for 3 s at 100 μl/min after each cycle. The kinetics of binding was analyzed by the Biacore T200 Evaluation software (GE Healthcare). The sensorgram for the binding experiment was normalized with the reference cell and fitted to the bivalent analyte model.

**High-speed centrifugation assay**. Various concentrations of Aip5 at the physiological buffer were incubated at room temperature for 30 min and then spun at 100,000 × g for 30 min. The supernatant fractions and pellet fractions were collected and analyzed by 10% SDS-PAGE electrophoresis and GelCode™ Blue Staining (also see Supplementary Fig. 10).

**Native gel analysis**. Four μg of purified Spa2-LC were incubated with 2 μg of Aip5 variants (Aip5-FL, Aip5-N, Aip5-N1014, Aip5-N457, and Aip5-N271), respectively, on ice for 30 min. The reactions were fractioned by 4% polyacrylamide stacking gels and 9% (w/v) polyacrylamide separating gels in 4 °C at 20 mV constant current for 2.5 h. The gel was stained by GelCode™ Blue Stain Reagent and destained by H$_2$O before scanning (also see Supplementary Fig. 10).

**Protein sequence prediction and conservative analysis**. To identify the Aip5 homologs and preform conservative analysis, the Aip5-C sequence was submitted as a query sequence in FungiDB (https://fungidb.org/fungidb/) with default parameters. The top 100 hits of Aip5 homologs were chosen from the species. Their correspondent sequence alignment was performed in the online server Clustal Omega (https://www.ebi.ac.uk/Tools/msa/clustalo/) and the phylogenetic tree was generated by the interactive tree of life (http://itol.embl.de/). ANCHOR (http://anchor.enzim.hu/) and PHYRE2 protein fold recognition server (http://www.sbg.bio.ic.ac.uk/~phyre/) were used, respectively, in identifying IDR and the conserved folded domains. Structural homologs of Aip5-C were identified from Dali server (http://ekhidna2.biocenter.helsinki.fi/dali/). Sequence alignment of Aip5-C and its structural homolog cGrx2 was carried out through T-COFFEE (http://tcoffee.crg.cat/) and subsequently generated by ESPript 3.0 (http://espript.ibcp.fr/ESPript/ESPript/).

**Fluorescent imaging analysis**. To analyze the signal intensity of Aip5-GFP at the presumptive bud site, the average Z-projection images were created from live cell imaging data by ImageJ. WT cells without florescence protein tagging were imaged at the same conditions to calculate the auto-florescence signal from the cytosol. The segmented line tool was used to draw a rectangular box with a width of 6 pixels and a length of 33 pixels at the bud tip perpendicular to the mother–daughter axis. The average intensity of the box was measured and termed as bud tip (T) signal intensity. The same size of the box was used in measuring the signal intensity of the cell cytosol region (C). The background region and auto-florescence signal from cytosol was measured in the same size box and termed as (B). The relative bud tip signal intensity to cytosol signal intensity (R) was calculated as $R = (T − B)/(C − B)$. The relative intensity profile R was classified into three categories: (1) no bud tip localization when $R = 1$; (2) a weak bud tip localization when $1 < R ≤ 1.5$; and (3) a strong bud tip localization when $R > 1.5$.

Living cell Abp140-3XGFP was imaged by z-axis scanning by wide-field microscope ×100 objective lens (NA 1.4), with a step size of 250 nm, which was close to the calculated ideal Nyquist Rate (https://svi.nl/NyquistCalculator). Image stacks were applied to the Huygens Essential deconvolution software (Scientific Volume Imaging, Netherlands) batch processing mode; a default calculation parameter was applied for fluorescence images to eliminate background signal and improve quality. The deconvoluted whole Z-scan images of Abp140-3XGFP were then subjected to the Imaris software version 8.0 (Bitplane, USA) and analyzed by FilamentTracer module to calculate the number, length, and total intensity of Abp140-labeled actin cables. The individual filament was tracked through the auto-tracked function with a threshold of 0.1 and filament thickness <0.5 μm. To analyze the lifetime of actin patches, time-lapse movies of Abp1-RFP were subjected to Huygens deconvolution using XYZT mode with a threshold of 0.1 and 10 calculations in the MLE algorithm. Kymographs were generated from movies of mother cells to measure the Abp1-RFP lifetime that reflects the endocytosis efficiency.

To analyze the in vivo Aip5-GFP aggregate formation under the stress conditions, wide-field microscopy Z-scan image series, at approximately 250 nm depth intervals, were subjected to Huygens deconvolution. Average signal Z-projection images were generated from 25 slices covering the entire cell volume. Trackmate function in Fuji ImageJ was used to recognize and quantify the in vivo aggregates. The aggregate intensity was measured by the Imaris spot detection module on the deconvoluted whole Z-scan image series.

**In vivo concentration measurements for polarisome proteins**. To quantify the in vivo protein concentration of polarisome proteins, three reference strains expressing Syp1-GFP, Crn1-GFP, and Sec3-GFP were used. The cytosolic protein concentration of Syp1, Crn1, and Sec3 were previously determined by FCCS[39,58]. Average intensity Z-projection images of polarisome protein tagged with the same GFP tag as reference proteins were used for intensity analysis. The total cytosol signal intensity (Ic) of reference and polarisome proteins were measured from the 10 × 10 pixel box in the cytoplasm of 20 cells. The background signal intensity and auto-fluorescence signal from the cytosol (Ib) were measured from 20 of 10 × 10 pixels areas from cells that did not express any fluorescence protein, as illustrated in Supplementary Fig. 5f. A signal intensity standard curve and equation were derived from the reported cytosolic concentration of reference proteins, Syp1, Crn1, and Sec3, and the measured value of Ic-Ib (Supplementary Fig. 5f). The cytosol concentrations of polarisome proteins were calculated based on the above equation. The protein concentration at the tip region was derived from the ratio of signal intensity (R). R was measured by the equation $R = (T − B)/(C − B)$, using the signal intensity in the background (B), bud tip (T), and cytosol (C) region. More than 20 medium-size budded cells were measured by ImageJ using a 3 × 35 pixel as ROI.

To validate the in vivo Aip5 concentration, we additionally quantified Aip5-GFP under endogenous promoter. Thirty ml of Aip5-GFP cells were cultured at the growing log phase from OD600 = 0.2 until 0.8 ($1.26 × 10^7$ cells) per ml and collected by centrifugation. The total cell proteins were extracted by TCA precipitation[57]. Three different cell count samples (OD600 = 1, 2, and 4, respectively) together with 3 samples of purified Aip5 recombinant proteins with a mass of 10.7, 21.4, and 42.8 ng, respectively, were loaded in the same SDS-PAGE gel. The detection of Aip5 protein amount was through western blot anti-GFP. And the individual protein band signal intensity was quantified in LI-COR system. The standard protein curve was generated by plotting the Aip5-GFP recombinant protein mass against the corresponding protein signal intensity. Thus the three-cell-count samples can derive the total protein mass from the standard curve. In order to calculate the Aip5 protein concentration in vivo, the protein mass (m: deducted from standard curve), Aip5 molecular weight (Mw = 143 kDa), cell number (n, calculated based on loaded OD600), and average haploid cell volume (V = 40 μm³) were taken into account based on the formula $C = m/Mw/n/V$.

**Statistical analysis**. All statistical analyses were performed in GraphPad Prism 6. p Values were determined by two-tailed Student's t test assuming equal variances and the one-way analysis of variance ($*p < 0.05$, $**p < 0.01$, $***p < 0.001$, $****p < 0.0001$, and ns = no significant). Error bars indicate the standard deviation (S.D.). The statistic source data are provided in Source Data file.

## Data availability

The coordinates and structure factors have been deposited in the Protein Data Bank with accession codes 6ABR (Aip5-C) and 6ABS (Aip5-C-3m). The source data underlying Figs. 1b, g, i, 2d, g, l, m, 3a, i, j, k, 4c, e, k, 5c, e, h, i, k, 6c–e and Supplementary Figs. 1j, 2c–e, g, i, 4e–f, 5f–g, i, 8e–g, i, j are provided as source data file. The materials and datasets generated during the current study are available from the corresponding author on reasonable request.

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

## Acknowledgements

We are grateful to Daniela Rhodes, Yasunori Saheki, and David Drubin for critical reading of the manuscript. We thank Jackie Tan for helping with the TEM work. We thank Min Wu (NUS, Singapore) for sharing the Nikon imaging system for TIRF microscopy. We thank David Drubin for sharing the strains. We also thank the NTU Protein Production Platform (www.proteins.sg) for protein expression test and purification of Spa2-C. This study was supported by NTU start-up grant (M4081533), MOE Tier 2 (MOE2016-T2-1-005S), Tier 1 (RG38/17-S), SRIS (SIG18002/M4062479) to Y.M., and MOE Tier 2 (MOE2015-T2-1-078) to Y.-G.G. in Singapore.

## Author contributions

Y.X., J.S., X.H., and A.T.-W. conducted the experiments. Y.X., J.S., A.T.-W., W.H., Y.-G. G., and Y.M. designed the experiments and analyzed the data. J.D.W.T. participated in the analysis of structural data. Y.X., J.S., Y.-G.G., and Y.M. wrote the manuscript, which all authors edited.

## Competing interests

The authors declare no competing interests.
