## [Peer Review File · Nature Communications]

Reviewers' comments:

Reviewer #1 (Remarks to the Author):

This manuscript by Xie and colleagues addresses a very interesting question: the possible role of low-complexity domains in proteins associated with actin assembly. They focus their attention on Aip5 protein, which they identified as a formin Bni1 interactor. They crystalize the only folded region of Aip5, its C-terminus which they show binds Bni1 and G-actin and promotes F-actin assembly in vitro. Aip5 N-terminus is of low-complexity and the authors show it forms aggregates in vitro, whose size is modulated by concentration and PEG in the buffer. Interestingly, this N-terminal part of Aip5 binds Spa2, which itself forms de-mixed liquid droplets in vitro and recruits Aip5 to these droplets. In vivo, Aip5 localizes to the bud tip dependent on Spa2, and *aip5* deletion shows synthetic sickness with *bni1Δ*. Upon energy depletion or lowering the pH, Aip5 forms aggregates in vivo, which are exacerbated in absence of Spa2.

Overall, the in vitro work is pretty impressive and complete, and demonstrates a very interesting property of Aip5 to form glass-like aggregates in absence of Spa2 but liquid droplets in presence of Spa2. The main unaddressed question is what the relevance of this behaviour is in vivo. Indeed, the cell biology work supports a role for Aip5 in promoting actin assembly, but the function of the aggregation and/or de-mixing is not directly addressed. A second issue is that the text needs clarification – there are many grammatical errors, which in several instances lead to possible misunderstanding or lack of clarity of the text. I address these issues in more detail below.

Link between in vitro observations and in vivo:

In general, the possible role in vivo of the phase separation observed in vitro is not addressed. In vitro, Spa2 is shown to recruit Aip5 to liquid droplets and it is proposed that the same happens in vivo upon stress (loosely defined), with Spa2 helping prevent Aip5 aggregates. The implication of this view is that Aip5 aggregates would be detrimental to the cell. However, this is not tested.

There are at least two testable predictions:

1. That the disorganization of actin observed in *spa2Δ* (fig 5g-h) depends on the formation of Aip5 aggregates: actin patches in *spa2Δ* upon energy depletion would be predicted to disappear upon deletion of Aip5;
2. That these changes have effect on cell viability in stressed conditions. Fig S6f shows that Aip5 C-terminus has a dominant negative effect at 37°C, which is not immediately concordant with the idea that Aip5 aggregates are detrimental. The authors should test whether *spa2Δ* has stress phenotypes, and whether deleting *aip5* alleviates these phenotypes.

More generally, the manuscript does not address at all what the function of the phase separation may be during normal growth. Addressing this would require the generation of Aip5 and/or Spa2 mutant with altered aggregation/droplet properties. Even if not addressed experimentally, it should be discussed. As is, the discussion is very superficial and vague and the model figure 6 does not propose any specific function for phase separation / aggregation by Spa2 and Aip5.

Role of Aip5:

I am not entirely convinced about the claimed synergy between Aip5 and Bni1 in actin assembly. The data clearly shows that Aip5 accelerates actin assembly on its own and does so also in presence of Bni1. The magnitude of the increase does not appear to be more with than without Bni1. The plots shown in fig 1d-e seem to indicate more additivity than synergy. In vitro, the only pieces of data truly convincing of synergy is the barbed end uncapping. Similarly, the genetic interaction between *aip5Δ* and *bni1Δ* (Fig 1h) suggest additive (independent) functions rather than Aip5 feeding into the Bni1 assembly function. This genetic interaction is however not entirely reproduced in Fig S6f. Thus, while the effect of Aip5 C-terminus on actin assembly is clear, the

possibly synergy with Bni1 should be phrased more carefully.

Similarly, the authors should be careful with their claimed *in vivo* Aip5 concentrations. Though GFP fluorescence can be used as a proxy to measure concentrations *in situ*, the uncertainty is significant. Claiming precise concentrations without standard deviation (which should be added) is misleading, especially as the difference in concentration measured at the cell tip and in the cytosol is less than 2-fold. As the authors point out, the actin polymerization activity of Aip5 is very modest at the measured *in vivo* concentrations. Given the formation of aggregates/liquid droplets, this measured concentration may be vastly off at the very local level. This would be worth discussing.

Text issues:

In general, the text needs attending to by a fluent English-speaker. I do not like to write this when it is purely esthetics, but there are instances here where the clarity of the text suffers (for instance, "Aip5 promotes actin assembly solely and synergistically with formin Bni1" is probably meant to state that Aip5 promotes actin assembly "on its own", as well as with Bni1, rather than "only with Bni1"; the sentence "In the absence or presence of crowded environments, Aip5 N-terminal variants showed amorphous assemblies in different size in the N-terminal length-dependent manner" is difficult to understand...). It would also help the reader to provide a little more interpretation to some of the results, as the text is rather dry.

The authors should also revise some of their description of the polarisome. For instance, the first sentence of the abstract sounds like the polarisome is both a well-established complex and an accepted form of non-membranous compartment, which has not been shown previously. The polarisome is a loosely-defined entity. To my knowledge, while individual components have been shown to interact with one another, there is no evidence that it forms a structural complex that can be purified as a whole (like the proteasome, the ribosome, ...). It is thus misleading to present it as such. I would also refrain from including Epo1 in it. At this rate any protein interacting with Bni1, Bud6 or Spa2 would be part of the polarisome, and this makes up dozens of proteins.

The discussion states "Aip5 that further recruits G-actin and Bni1", but I do not think the paper shows that Aip5 recruits Bni1.

Other comments:

The text in Fig S1A is so small that it is very difficult to read.

In fig 1f-g, the concentration of proteins used is not indicated.

In Fig 2g-h, the Aip5C alone controls are missing.

Fig 5h legend does not state what condition is quantified. In either case, it is missing either the energy depletion of the control to test whether the effect is specific to energy depletion or whether *spa2Δ* has an effect also in steady-state growth.

I do not understand the statement that "full length Aip5 eluted earlier than the predicted size on gel filtration column chromatography (Fig S4A)". What is the predicted size, and at what size does it elute?

The effect of L-arginine on the elution of Aip5 is not very clear.

Reviewer #2 (Remarks to the Author):

Xie et al discovered a new member in polarisome, a multiprotein center for actin nucleation in budding yeast and filamentous fungi. The authors named the new member Aip5 and carried out detailed characterization for Aip5 with various assays. Two major conclusions are made: (I) C-terminus folded domain of Aip5 promotes actin nucleation by interacting with G-actin and Bni1; (II) N-terminus disordered domain of Aip5 forms functional high-order oligomer, undergoes condensation when crowded, and demixes with Spa2 for stress adaption. I found conclusion (I) is well supported with data, but not so with conclusion (II). I would not recommend for publication unless the authors address my following concerns.

1. the data does not support the notion that Aip5 itself undergo liquid condensation (Figure 4d).
2. The authors show Aip5 is a client molecule that can be recruited to Spa2 condensates. They looked at the effect on Aip5 dynamics, they should also look at how this affects actin nucleation by Aip5. Without this data, the title is not meaningful, then the authors should change the title.
3. The authors argue that interacting with Spa2 helps Aip5 stress adaption by looking at Aip5 'aggregates' under stress conditions and Spa2 deletion background. Authors did no characterization with Aip5 puncta in stressed cells and yet implied that they are in similar aggregating state within that in vitro (Figure 4d which the authors called condensates instead of aggregates, without any justification). Based on the reversibility and sensitivity to 1,6 hexanediol, Aip5 assembly in stressed cells represent liquid condensates. The authors should further characterize the material properties for Aip5 assembly in vitro and in stressed cells before implying they are the same. One alternative explanation for the observations under stress condition is Aip5 goes to liquid stress bodies and Spa2 has an effect on the formation of such stress body. To rule out this possibility, the authors should also look at how Spa2 condensation and the recruitment of Aip5 to Spa2 condensates are affected by pH in vitro, and also characterize Spa2 localization in stressed cells, its relation to Aip5 foci and the material properties.
4. The authors made a lot of assumptions when using different terms to describe material properties of protein assemblies. For example, Line 194: coalescence. No data is given to shown the growth is via coalescence. Line 229: glassy, the FRAP data only show it's not dynamic, no data is given to show the material is a glass.

Reviewers' comments:

Reviewer #1 (Remarks to the Author):

This manuscript by Xie and colleagues addresses a very interesting question: the possible role of low-complexity domains in proteins associated with actin assembly. They focus their attention on Aip5 protein, which they identified as a formin Bni1 interactor. They crystalize the only folded region of Aip5, its C-terminus which they show binds Bni1 and G-actin and promotes F-actin assembly *in vitro*. Aip5 N-terminus is of low-complexity and the authors show it forms aggregates *in vitro*, whose size is modulated by concentration and PEG in the buffer. Interestingly, this N-terminal part of Aip5 binds Spa2, which itself forms de-mixed liquid droplets *in vitro* and recruits Aip5 to these droplets. *In vivo*, Aip5 localizes to the bud tip dependent on Spa2, and *aip5* deletion shows synthetic sickness with *bni1Δ*. Upon energy depletion or lowering the pH, Aip5 forms aggregates *in vivo*, which are exacerbated in absence of Spa2.

Overall, the *in vitro* work is pretty impressive and complete, and demonstrates a very interesting property of Aip5 to form glass-like aggregates in absence of Spa2 but liquid droplets in presence of Spa2. The main unaddressed question is what the relevance of this behaviour is *in vivo*. Indeed, the cell biology work supports a role for Aip5 in promoting actin assembly, but the function of the aggregation and/or de-mixing is not directly addressed. A second issue is that the text needs clarification – there are many grammatical errors, which in several instances lead to possible mis-understanding or lack of clarity of the text. I address these issues in more detail below.

Link between *in vitro* observations and *in vivo*:

In general, the possible role *in vivo* of the phase separation observed *in vitro* is not addressed. *In vitro*, Spa2 is shown to recruit Aip5 to liquid droplets and it is proposed that the same happens *in vivo* upon stress (loosely defined), with Spa2 helping prevent Aip5 aggregates. The implication of this view is that Aip5 aggregates would be detrimental to the cell. However, this is not tested. There are at least two testable predictions:

1. That the disorganization of actin observed in *spa2Δ* (fig 5g-h) depends on the formation of Aip5 aggregates: actin patches in *spa2Δ* upon energy depletion would be predicted to disappear upon deletion of Aip5;

We thank Reviewer #1 for this great suggestion to elucidate the function of Aip5 aggregates *in vivo* better. We have now added the following experiments to discuss this point.

First, upon energy depletion, the disorganization of actin cable starts as early as 15 min in wild type and around 5 min in the *aip5Δ* and *spa2Δ* mutant cells (Figure S9a). This data suggests that both Spa2 and Aip5 involve in the cellular protection of actin cables under stress conditions. Consistently, deletion of *AIP5* in the WT or in the background of *spa2Δ* did not reduce actin patches upon energy depletion (Figure S9a).

Second, the Aip5 aggregates appeared starting from 30 min (Figure S9b), which is later than the actin cable disorganization starting from ~ 15min (Figure S9b). It indicates that the actin remodeling within the initial 30 min upon energy depletion is less likely due to Aip5 aggregation.

We discussed these new results in the last paragraph of the Discussion section. We think the formation of the aggregates of Aip5 is a cellular response to energy depletion from the nature of its intermolecular multivalency that was used as a temporary and rapid solution in response to stress conditions (Figure 6a).

2. That these changes have effect on cell viability in stressed conditions. Fig S6f shows that Aip5 C-terminus has a dominant negative effect at 37°C, which is not immediately concordant with the idea that Aip5 aggregates are detrimental. The authors should test whether *spa2Δ* has stress phenotypes, and whether deleting *aip5* alleviates these phenotypes.

We appreciate Reviewer#1 for raising the excellent point. We have carried out a new experiment and added it in Figure 6g now. Under the prolonged adverse condition, we found that Aip5 aggregates are indeed detrimental to the cell as Reviewer #1 predicted. Once we deleted *AIP5* in the background of *spa2Δ*, it alleviated the cell growth sickness comparing to *spa2Δ* mutant cells. It indicates Aip5 *in vivo* aggregates are toxic under long-term adverse conditions at the later stage of stress. We have now added the new data in Figure 6g and discussed the function Aip5 aggregates carefully in the last paragraph of the Discussion.

In addition, to better define Spa2 condensates recruits Aip5 *in vivo*, we added new experiments that showed Aip5 demixes with Spa2 that controls the size and structure of *in vivo* Spa2-Aip5 macromolecular condensates (Figs. 5d-h and s8a, d-i). Aip5 condensates undergo rapid recovery once favourable nutrients were supplemented. In contrast, without Spa2 demixing effect, Aip5 condensates has slow recovery *in vivo*, and also detrimental to cell survival if under long-term stress. Together, these observations agreed with *in vitro* data where Spa2-Aip5 droplet reversed Aip5 amorphous assemblies by increasing dynamics of Aip5.

More generally, the manuscript does not address at all what the function of the phase separation may be during normal growth. Addressing this would require the generation of Aip5 and/or Spa2 mutant with altered aggregation/droplet properties. Even if not addressed experimentally, it should be discussed. As is, the discussion is very superficial and vague and the model figure 6 does not propose any specific function for phase separation/aggregation by Spa2 and Aip5.

We thank Reviewer#1 great suggestion and apologize for the lack of discussion about the physiological function of Aip5/Spa2 phase separation.

First, as Reviewer #1 suggested, we have performed new experiments and discussed more in the first paragraph of the Discussion section.

Second, to investigate more the mechanisms by which the Aip5 multivalency, including the dimerization and oligomerization states, directly regulate its activities as nucleator and NPF of Bni1, we

performed the following additional experiments. 1, we compared Aip5 activities between the Aip5 dimer and Aip5 monomer. We generated a monomeric version of Aip5-C by truncating away 35 amino acids from its N terminal region (Figures s4a-c). Compared to Aip5-C dimer, the monomeric version of Aip5-C (Aip5-mC) showed an apparent reduction in biochemical activities for assembling actin and promoting Bni1-mediated nucleation (Figures s4d-f), suggesting the importance of the dimeric interaction of Aip5 for its biochemical activity. 2, to determine the changes of Aip5 activities by a high-ordered oligomerization state through Spa2 recruitment, we examined Aip5-mediated actin polymerization rate, in the absence or presence of Spa2-LC, with or without crowding agent (Figure R1). Unfortunately, in such *in vitro* condition, we did not observe the significant change of the Aip5 activities from the enhanced multivalent interaction with Spa2. At this moment, without a full reconstitution of polarisome complex proteins, including Bni1, Spa2, Aip5, and Bud6, which comprise all necessary interactions domains for orchestrating a multivalent complex, we would not explicitly claim the regulatory mechanism of Aip5 activities from its demixing into Spa2 droplet. This comment is in the same line as Reviewer #2. To prevent any potential misleading at this stage, as Reviewer #2 suggested, we changed the title to “Macromolecular condensation of actin assembly factor Aip5 in polarisome complex”. We wish Reviewer #1 endorse our opinion that such title change does not affect the quality, the conclusion, and the significance of our report.

Furthermore, as suggested, we also amended our model in Figure 7 to indicate the different functions and states of Aip5 assemblies under normal growth conditions and stress conditions.

Figure R1. The relative actin assembly rate of Aip5 at the indicated concentrations where in combination with/without 5 μM Spa2-LC . And 5% PEG3350 was used to create the crowded environment for actin assembly. All the data points were generated from 3 biological replicates. The error bar represents S.D.

Role of Aip5:

I am not entirely convinced about the claimed synergy between Aip5 and Bni1 in actin assembly. The data clearly shows that Aip5 accelerates actin assembly on its own and does so also in presence of Bni1. The magnitude of the increase does not appear to be more with than without Bni1. The plots shown in fig 1d-e seem to indicate more additivity than synergy. *In vitro*, the only pieces of data truly convincing of synergy is the barbed end uncapping. Similarly, the genetic interaction between *aip5Δ* and *bni1Δ* (Fig

1h) suggest additive (independent) functions rather than Aip1 feeding into the Bni1 assembly function. This genetic interaction is however not entirely reproduced in Fig S6f. Thus, while the effect of Aip5 C-terminus on actin assembly is clear, the possibly synergy with Bni1 should be phrased more carefully.

We have now performed additional biochemistry experiments using total internal reflection fluorescence microscopy (TIRFM) assay to strengthen our results and rephrased our statement carefully regarding the synergy effect between Aip5 and Bni1 *in vivo*.

First, we performed more *in vitro* biochemical experiments to quantitatively analyze Aip5 activities. The bulk actin assembly assay pyrene assay (Figures 1d, e) on its own could not distinguish Aip5's function in different events of actin assembly, such as actin nucleation, elongation or even actin filament annealing. To provide better and quantitative biochemical analysis for *in vitro* synergy activities between Aip5 and Bni1, we have now performed additional TIRF experiment to monitor actin nucleation and elongation in real-time (Figures 1f-i), which can better dissect the actin assembly activities, for both nucleation and elongation. From the new TIRF experiment, we quantified the actin nucleation seeds number at early time point 5 min as well as actin filament barbed end elongation speed. A significant increase of actin seeds number was observed when introducing both Aip5 and Bni1 in the same reaction, an average of 99 actin seeds were formed in the defined areas for quantification, which is greater than the sum of actin seeds (42 actin seeds) that are generated by Aip5 and Bni1 on their own. Furthermore, neither Aip5 nor Bni1 affected actin filament barbed end elongation speed. We hope Reviewer #1 finds our new data supports the synergistic effect in actin nucleation better now.

Second, we agree with Reviewer #1's comment on the growth results of the double mutant. Since the growth assay is not as quantitative as the biochemical assay and may reflect a mixed effect, the more growth defects of double mutant than the single mutant could not distinguish the additive (independent) and synergistic function of BNI1 and AIP5 easily. We think our new TIRF assay offers more validation now that Aip5 has both nucleation and NPF activities, though Aip5 could also have additional functions in the polarisome complex. The conclusion about the NPF activity of Aip5 in an exclusive manner by using the growth assay results of double mutant *aip5Δ* and *bni1Δ* is not appropriated. We have now rephrased the sentence and described the result more carefully.

Similarly, the authors should be careful with their claimed *in vivo* Aip5 concentrations. Though GFP fluorescence can be used as a proxy to measure concentrations *in situ*, the uncertainty is significant. Claiming precise concentrations without standard deviation (which should be added) is misleading, especially as the difference in concentration measured at the cell tip and in the cytosol is less than 2-fold. As the authors point out, the actin polymerization activity of Aip5 is very modest at the measured *in vivo* concentrations. Given the formation of aggregates/liquid droplets, this measured concentration may be vastly off at the very local level. This would be worth discussing.

We thank Reviewer#1 for the concern of concentration measurement and suggestions. We did the following new experiments to address these concerns.

First, we apologize for the missing SD value, which is corrected now in Figure S5g.

Second, to validate the fluorescent image-based approach in measuring Aip5 *in vivo* concentrations, we performed additional western blot assay. The new results are added in Figures S5h, i. We generated a standard curve using recombinant Aip5-GFP protein by western blot detection via anti-GFP, where the standard curve was plotted from Aip5-GFP loading mass against the quantified protein signal intensity from western blot. *In vivo* Aip5-GFP concentration were estimated by detect endogenous Aip5-GFP expressing from the total cell lysate. In order to calculate the Aip5 protein concentration *in vivo*, the protein mass (m: deducted from standard curve), molecular weight (Mw=143 kDa), cell number (n, calculated based on loaded O.D.600) and average haploid cell volume (V=40 μm^3) were taken into account based on the formula $C=m/Mw/n/V$.

From the western blot approach, we found that the calculated Aip5 protein concentration through western blot (~80 nM) was similar to that was derived from fluorescence method (~83 nM in the cytoplasm). By considering the results from both experiments, we think our results showed a general protein concentration of Aip5 *in vivo*. However, we do acknowledge Reviewer #1's concern about the potential inaccuracy of the exact protein concentration because both western-blot and microscopic approaches have their technical limitations. Fluorescence (cross)-correlation spectroscopy used by Kaksonen and Knop group (Boeke et al., 2014) could be a more accurate approach for *in vivo* protein measurement, especially for low abundant proteins. Here, we added new western-blot result, and also described the potential limitation for current protein concentration measurement in the Discussion section. We found that our presented protein concentration is useful in addressing Reviewer #1's other concerns.

- a) In the initially submitted manuscript, we used pyrene actin polymerization assay, which is not sensitive enough to demonstrate the Aip5 functions as nucleator and NPF of Bni1. Now, in the revised Figures 1f-g using TIRF actin assay, compared to the reaction without Aip5, 20 nM of Aip5 is sufficient to generate more actin seeds, during both the spontaneous actin polymerization and synergistic nucleation in Bni1-mediated actin assembly. Therefore, such *in vivo* low Aip5 concentration (<100nM) is sufficient to provide biochemical activities in actin assembly.
- b) Such protein concentration measurement could provide a proximate stoichiometry of Aip5 and Spa2 that guided us to demonstrate *in vitro* Aip5 participation in Spa2 droplet at a concentration and stoichiometry close to *in vivo* situation (Figure S7f). We demixed Spa2-LC and Aip5 at a stoichiometry of ~5:1 with a concentration of 500 nM and 100 nM, respectively, which is close to their physiological concentration and the relative stoichiometry at bud tip. In results, we could still observe the demixing of Aip5 into the Spa2-LC droplet in the presence of crowding reagent. Now we added this new result to Figures S7f and more discussion.

Text issues:

In general, the text needs attending to by a fluent English-speaker. I do not like to write this when it is purely esthetics, but there are instances here where the clarity of the text suffers (for instance, "Aip5 promotes actin assembly solely and synergistically with formin Bni1" is probably meant to state that Aip5 promotes actin assembly "on its own", as well as with Bni1, rather than "only with Bni1"; the sentence "In the absence or presence of crowded environments, Aip5 N-terminal variants showed

amorphous assemblies in different size in the N-terminal length-dependent manner" is difficult to understand...). It would also help the reader to provide a little more interpretation to some of the results, as the text is rather dry.

We apologize for the confusion, we have now revised it and provided more interpretation for the results.

The authors should also revise some of their description of the polarisome. For instance, the first sentence of the abstract sounds like the polarisome is both a well-established complex and an accepted form of non-membranous compartment, which has not been shown previously. The polarisome is a loosely-defined entity. To my knowledge, while individual components have been shown to interact with one another, there is no evidence that it forms a structural complex that can be purified as a whole (like the proteasome, the ribosome, ...). It is thus misleading to present it as such. I would also refrain from including Epo1 in it. At this rate any protein interacting with Bni1, Bud6 or Spa2 would be part of the polarisome, and this makes up dozens of proteins.

We thank Reviwer#1 for these excellent comments. We have now revised our description in the Abstract.

The discussion states "Aip5 that further recruits G-actin and Bni1", but I do not think the paper shows that Aip5 recruits Bni1.

We have now corrected it to a more precise way "Aip5 that directly interacts with G-actin and Bni1".

Other comments:

The text in Fig S1A is so small that it is very difficult to read.

We have now enlarged the font size for better visualization.

In fig 1f-g, the concentration of proteins used is not indicated.

Thanks for pointing this out. We added the protein concentration now in the figure legend.

In Fig 2g-h, the Aip5C alone controls are missing.

To improve the quality and clarity, we have now performed additional actin polymerization experiments by monitoring actin nucleation using total internal reflection fluorescence (TIRF) microscopy. All the control and quantification are added now in the revised Figures 2f and g.

Fig 5h legend does not state what condition is quantified. In either case, it is missing either the energy depletion of the control to test whether the effect is specific to energy depletion or whether *spa2Δ* has an effect also in steady-state growth.

We apologize for the missing text and quantification graph. We have now specified the conditions for quantification in Figures 6c-e. We have also now quantified the control groups and added in Figure S8j and amended the figure legends accordingly. Compared to WT cells, there is a mild defect with shorter actin filament in *spa2Δ* at the steady-state growth, which is added now in Figure S8j.

I do not understand the statement that "full-length Aip5 eluted earlier than the predicted size on gel filtration column chromatography (Fig S4A)". What is the predicted size, and at what size does it elute?

To clarify the confusion, now we added the expected elution volume of Aip5 dimers in the revised Figure S5a. The predicted elution volume from a gel filtration column (Superdex™ 200 Increase column) based on Aip5 dimer molecular weight (286 kDa) would be around 11-13 ml. However, our purified Aip5 proteins usually eluted from 8-10 ml (larger than 440 kDa), suggesting a higher-order of oligomerization.

The effect of L-arginine on the elution of Aip5 is not very clear.

We added more results to clarify this point now. 1st, EM negative staining, we found L-arginine break down the Aip5 oligomers from large assemblies (Figure 4a) into a dissolved pattern in Figure S5c. 2nd, size exclusion chromatography (SEC) did not show a dramatic shift of Aip5 elution peak when treated with L-arginine (Figure S5a), but with shifted tail lagging behind during elution. It is a typical resolution that SEC can resolve that indicate significant dissolution of large size oligomer (Figure S5c). Our results indicate a disruption of Aip5 large assemblies by L-arginine into a lower-order of Aip5-oligomer but were not sufficient to be in its monomeric state. This L-arginine effect in elution of Aip5 oligomers was similar to our previously reported profilin oligomerization, in the presence or absence of L-arginine, which is shown in the Figure 2k and 2L of Sun He et. al. (Sun et al., 2018) (also shown below). With L-Arginine, the oligomerization of high-order Arabidopsis profilin 3 is reduced (a shift of early elution peak towards right), but not to the level of monomer completely. Now, we revised the results with clearer description in the Page 4 line 209-212.

(K) Reduced AtPRF3 oligomerization by applying 0.2 M L-Arginine. (L) Oligomerization of AtPRF3 variants on the SDS-PAGE gel and visualized by silver staining. 50 μ M of each AtPRF3 variant was incubated with 100 μ M PolyP and cross-linked with 1.5 mM MTS-2-MTS for 1 hour at room temperature. (Sun He et. al.)

Reviewer #2 (Remarks to the Author):

Xie et al discovered a new member in polarisome, a multiprotein center for actin nucleation in budding yeast and filamentous fungi. The authors named the new member Aip5 and carried out detailed characterization for Aip5 with various assays. Two major conclusions are made: (I) C-terminus folded domain of Aip5 promotes actin nucleation by interacting with G-actin and Bni1; (II) N-terminus disordered domain of Aip5 forms functional high-order oligomer, undergoes condensation when crowded, and demixes with Spa2 for stress adaption. I found conclusion (I) is well supported with data, but not so with conclusion (II). I would not recommend for publication unless the authors address my following concerns.

1. the data does not support the notion that Aip5 itself undergo liquid condensation (Figure 4d).

We agree with Reviewer #2. We did not state that Aip5 undergoes LLPS on its own. Aip5 forms high-order oligomers under the crowding condition, which showed low motility. And it only becomes more fluidic by demixing with the liquid droplet of Spa2 *in vitro* (Figures 4j.k). Furthermore, we have now examined the dynamic of Aip5's polarized tip localization, as well as the small foci formed in wild type and *spa2Δ* under energy depletion conditions *in vivo* (Figures 5d-f). We observed the full recovery of Aip5 protein after photobleaching at the polarized tip region (within 20 sec) where it bound to various polarisome components, Spa2/Bni1. On the contrary, the foci formed in *spa2Δ* under energy depletion has shown worse dynamic property comparing to the foci formed in wild type, which agreed with the notion that *in vitro* Aip5 protein phase separated as less-mobile aggregates without its binding partner Spa2. To avoid any potential misleading information, we have now double confirmed our description and phrased them carefully in the revised manuscript.

2. The authors show Aip5 is a client molecule that can be recruited to Spa2 condensates. They looked at the effect on Aip5 dynamics, they should also look at how this affects actin nucleation by Aip5. Without this data, the title is not meaningful, then the authors should change the title.

We appreciate this excellent point from Reviewer #2.

To study the mechanisms by which the Aip5 multivalency, including the dimerization and oligomerization states, directly regulates its activities as nucleator and NPF of Bni1, we performed the following experiments. Firstly, we compared Aip5 activities between the Aip5 dimer and Aip5 monomer. We generated a monomeric version of Aip5-C by truncating away 35 amino acids from its N terminal region (Figures s4a-c). Compared to Aip5-C dimer, the monomeric version of Aip5-C (Aip5-mC) showed a noticeable reduction in biochemical activities for assembling actin and promoting Bni1-mediated nucleation (Figures s4d-f), suggesting the importance of the dimeric interaction of Aip5 for its biochemical activity. Secondly, to determine the changes of Aip5 activities by a high-ordered oligomerization state through Spa2 recruitment, we examined Aip5-mediated actin polymerization rate, in the absence or presence of Spa2-LC, with or without crowding agent (Figure R1). Unfortunately, we did not observe the significant change of the Aip5 activities from the enhanced multivalent interaction with Spa2. At this moment, without a full reconstitution of polarisome complex proteins, including Bni1, Spa2, Aip6, and Bud6, which comprise all necessary interactions domains for orchestrating a multivalent

complex, we would not explicitly claim the regulatory mechanism of Aip5 activities by demixing into Spa2 droplet.

As suggested, we changed the title to “Macromolecular condensation of actin assembly factor Aip5 in polarisome complex”. We also added these discussions in the first paragraph of the Discussion section. We wish Reviewer #2 endorses our opinion that such title change does not affect the quality, the conclusion, and the significance of our report.

3. The authors argue that interacting with Spa2 helps Aip5 stress adaption by looking at Aip5 ‘aggregates’ under stress conditions and Spa2 deletion background. Authors did no characterization with Aip5 puncta in stressed cells and yet implied that they are in similar aggregating state within that *in vitro* (Figure 4d which the authors called condensates instead of aggregates, without any justification). Based on the reversibility and sensitivity to 1,6 hexanediol, Aip5 assembly in stressed cells represent liquid condensates. The authors should further characterize the material properties for Aip5 assembly *in vitro* and in stressed cells before implying they are the same. One alternative explanation for the observations under stress condition is Aip5 goes to liquid stress bodies and Spa2 has an effect on the formation of such stress body. To rule out this possibility, the authors should also look at how Spa2 condensation and the recruitment of Aip5 to Spa2 condensates are affected by pH *in vitro*, and also characterize Spa2 localization in stressed cells, its relation to Aip5 foci and the material properties.

We appreciate Reviewer #2’s for the constructive suggestions and apologize for the missed justification. We have now added the suggested experiments and with a better description, explanation, and discussion.

Firstly, to better understand the *in vivo* Aip5 foci property under stressed conditions, we applied FRAP to examine the protein dynamic (Figures 5d, e). Indeed, the result agreed with 1,6 hexanediol observation where Aip5 foci formed in wild type dissolved earlier than those formed in *spa2Δ* (Figure s8a), which Aip5 foci recovered faster and better with Spa2 protein within the cells. Furthermore, the rapid reversal of *in vivo* Aip5 condensates (around 50% recovery within 1 min, Figures 5d,e) and the slow FRAP recovery of *in vitro* Aip5 assembly on its own (less than 20% recovery within 4 min, Figures. 4j, k) suggest that the dynamic behaviors of pure recombinant Aip5 is not identical to the *in vivo* Aip5 assemblies. One possibility for the different behavior between *in vitro* and *in vivo* of Aip5 is that Aip5 has other binding partners that help Aip5 in a better fluidic state, even in the *spa2Δ* strain. We have demonstrated that Bni1 (Figures S1c-f; Figures 2i-m) is the binding partner of Aip5 which affected the polarized Aip5 tip localization *in vivo*. On the other hand, deletion of Bud6 diminished Aip5 tip localization, implying the Bud6 involvement in partial recruitment of Aip5 *in vivo* (Figures 1a, b). In addition, when Aip5 localized to the polarized tip where it binds to various polarisome proteins, Aip5 showed the most dynamic behavior from FRAP experiment (Figures 5d, e). Taken together, our newly added FRAP assay and 1,6 hexanediol suggested Aip5 exhibits the properties expected for biomolecular

liquid-like condensates *in vivo*. Similar characterizations were also reported for liquid-like condensates in nuclear (Sabari et al., 2018).

Secondly, we performed the suggested *in vitro* pH experiment for Spa2-Aip5. The new results are added in Figures S8c-g. We found Spa2-LC still formed droplet at pH range 4.4-7.5, though some droplets with irregular shape were observed at pH4.4 (Figure S8c). We next tested the recruitment of Aip5 into condensates Spa2-LC at physiological pH conditions. We chose the same pH conditions as we used for *in vivo* stress treatment in Figures 5j,k. Though Aip5 can be recruited into Spa2-LC droplet under all the three pH conditions, the Aip5 dynamics in each droplet are different (Figures S8d-g). At pH5.5, demixing with Spa2-LC did not improve Aip5 motility. On the other hand, Aip5 showed the better fluidic property in Spa2-droplets at both pH6.5 and pH7.5 when compared to Aip5 assemblies without Spa2. This new *in vitro* result was consistent with the *in vivo* observation that Aip5 formed more pronounced condensates at pH5.5 than under pH6.5/pH7.5 in wild type cells (Figures 5j,k). It indicates pH is another regulator factor for Aip5 phase behavior. Under a low pH of 5.5, a presence of Spa2 could not alleviate the decreased motility of Aip5 assemblies. We further confirmed *in vitro* pH-dependent experiment *in vivo* by examining Aip5 dynamic using FRAP at pH 5.5 (Figures S8h, i), where there was also no significant difference in the recovery rate of Aip5-GFP in wild type and *spa2Δ in vivo*. In contrast, the Aip5 foci formed in *spa2Δ* at pH6.5 recovered faster than that under pH5.5, both *in vivo* and *in vitro* (Figures S8f,h, i). Above data also suggest that other cellular components besides Spa2 play roles in regulating the dynamics of Aip5 *in vivo* condensates. Aip5 condensates in the SPA2 were barely detectable under pH6.5 and pH7.5. Thus, we were not able to examine their dynamic recovery rates through FRAP.

Third, *in vivo* Spa2 localization experiment is an excellent suggestion. We have examined Spa2 localization together with Aip5 during stress treatment. a), under energy depletion condition, Aip5 condensates highly colocalized with Spa2 (Figure 5f), which is consistent with the function of Spa2 in regulating Aip5 motility (Figures 5d, e). b), the Spa2 condensate formed during energy depletion condition also displayed a recovery property similar to Aip5, both of which recovered to around 50% in 60 seconds after photo-bleaching (Figures 5g, h). c), [redacted]

[redacted]

4. The authors made a lot of assumptions when using different terms to describe material properties of protein assemblies. For example, Line 194: coalescence. No data is given to show the growth is via coalescence. Line 229: glassy, the FRAP data only show it's not dynamic, no data is given to show the material is a glass.

First, we appreciate Reviewer #2's for pointing out the inappropriate word "coalescence" we used. We have removed this word. Second, we agree that we need to be careful about the word "glassy" state. In the previous report from Rohit V. Pappu group, glassy intra-condensates are used to describe the condensates that have a lag time for disassembly (Holehouse and Pappu, 2018). In material science, the term "glass" is restricted to the properties of a formation of amorphous material through a decrease of temperature, and a reversible transition by a temperature increase. We do not have such physical-chemical characterization approaches to determine the material properties, such as calorimetric ideal glass transition temperature T_{0c} , etc. Now to prevent any misleading, we have now revised the term "glassy" state to "less-motile assemblies".

Boeke, D., Trautmann, S., Meurer, M., Wachsmuth, M., Godlee, C., Knop, M., and Kaksonen, M. (2014). Quantification of cytosolic interactions identifies Ede1 oligomers as key organizers of endocytosis. *Mol Syst Biol* 10, 756.

Holehouse, A.S., and Pappu, R.V. (2018). Functional Implications of Intracellular Phase Transitions. *Biochemistry* 57, 2415-2423.

Sabari, B.R., Dall'Agnesse, A., Boija, A., Klein, I.A., Coffey, E.L., Shrinivas, K., Abraham, B.J., Hannett, N.M., Zamudio, A.V., Manteiga, J.C., *et al.* (2018). Coactivator condensation at super-enhancers links phase separation and gene control. *Science* 361.

Sun, H., Qiao, Z., Chua, K.P., Tursic, A., Liu, X., Gao, Y.G., Mu, Y., Hou, X., and Miao, Y. (2018). Profilin Negatively Regulates Formin-Mediated Actin Assembly to Modulate PAMP-Triggered Plant Immunity. *Curr Biol* 28, 1882-1895 e1887.

Reviewer #1 (Remarks to the Author):

In this revised manuscript, the authors have performed substantial new experiments that address most of the concerns I had raised. The link between the in vitro data and the situation in vivo is better documented. I support publication of this manuscript in Nature Communications. I would just suggest the authors address by minor text changes the three following points.

The authors may want to slightly re-phrase this following sentence because 0.1 to 2 μ M does NOT cover the measured 80nM in vivo (0.1 μ M = 100 nM): "The examined range of Aip5 concentration from 0.1 to 2 μ M covers the physiological intracellular concentration of Aip5, which showed to be ~83 nM in vivo using microscopic imaging approach and ~80 nM using western-blot based assay (Supplementary Figs. 5f-i)." I am quite ready to believe that the local concentration of Aip5 at the bud tip will be higher than the global cellular concentrations measured, but this needs to be explicitly stated.

I am slightly confused by the apparent formation of aggregates by the Aip5C fragment in vivo upon energy depletion (in Fig 5b). The authors may want to add a sentence explaining this.

The authors have added new data that Aip5 assemblies in vivo are dissolved by 1,6 hexanediol (Fig S8a). However, 1,6 hexanediol appears not to dissolve the Aip5 assemblies in spa2 Δ cells. This piece of data should be explicitly mentioned in the text.

Reviewer #2 (Remarks to the Author):

The added data clarified Spa2-mediated Aip5 phase behavior in normal and stressed conditions and its roles in actin assembly. This is described well in the new model in Figure 7. I appreciate the authors' effort to adapt the title based on the new finding that Spa2-mediated Aip5 phase separation does not promote actin assembly in vitro. I think the original title is fine because now the authors have added more data and discussion. Plus the current title does not convey key messages of the paper. I hope the authors can also adapt the abstract based on the new data by adding a description of roles of Spa2-mediated Aip5 phase separation in actin assembly in normal and stressed cells, possibly before the last sentence. I still find the terms the authors used to describe Aip5 assembly in vitro and in vivo confusing: I've seen condensation/condensates, phase separation, aggregate, amorphous assembly, reversible aggregates and cellular bodies. Reading the previous version I had the sense the authors were using them interchangeably and implying they were the same. Since the authors have added new data showing there is a difference in Aip5 assembly in different conditions, the authors should just choose a name for each situation, define it and use it throughout the whole paper to avoid confusion.

REVIEWERS' COMMENTS:

Reviewer #1 (Remarks to the Author):

In this revised manuscript, the authors have performed substantial new experiments that address most of the concerns I had raised. The link between the *in vitro* data and the situation *in vivo* is better documented. I support publication of this manuscript in Nature Communications. I would just suggest the authors address by minor text changes the three following points.

The authors may want to slightly re-phrase this following sentence because 0.1 to 2 μ M does NOT cover the measured 80nM *in vivo* (0.1 μ M = 100 nM): "The examined range of Aip5 concentration from 0.1 to 2 μ M covers the physiological intracellular concentration of Aip5, which showed to be ~83 nM *in vivo* using microscopic imaging approach and ~80 nM using western-blot based assay (Supplementary Figs. 5f-i)." I am quite ready to believe that the local concentration of Aip5 at the bud tip will be higher than the global cellular concentrations measured, but this needs to be explicitly stated.

We appreciate reviewer #1 suggestion. Now we have corrected the sentence to be more clearly deliver the idea that the bud tip concentration of Aip5 is higher than the average cytosolic protein concentration.

I am slightly confused by the apparent formation of aggregates by the Aip5C fragment *in vivo* upon energy depletion (in Fig 5b). The authors may want to add a sentence explaining this.

Thanks reviewer #1 for pointing out this point, currently we have added one sentence to interpret this result.

The authors have added new data that Aip5 assemblies *in vivo* are dissolved by 1,6 hexanediol (Fig S8a). However, 1,6 hexanediol appears not to dissolve the Aip5 assemblies in *spa2* Δ cells. This piece of data should be explicitly mentioned in the text.

We appreciate reviewer #1 suggestion, now we have added on more explanation to the 1,6 hexanediol data and defined different state of Aip5 foci based on the response to 1,6 hexanediol treatment under energy depletion condition.

Reviewer #2 (Remarks to the Author):

The added data clarified Spa2-mediated Aip5 phase behavior in normal and stressed conditions and its roles in actin assembly. This is described well in the new model in Figure 7. I appreciate the authors' effort to adapt the title based on the new finding that Spa2-mediated Aip5 phase separation does not promote actin assembly *in vitro*. I think the original title is fine because now the authors have added more data and discussion. Plus the current title does not convey key messages of the paper. I hope the authors can also adapt the abstract based on the new data by adding a description of roles of Spa2-mediated Aip5 phase separation in actin assembly in normal and stressed cells, possibly before the last sentence. I still find the terms the authors used to describe Aip5 assembly *in vitro* and *in vivo* confusing: I've seen condensation/condensates, phase separation, aggregate, amorphous assembly, reversible aggregates and cellular bodies. Reading the previous version I had the sense the authors were using them interchangeably and implying they were the same. Since the authors have added new data showing there is a difference in Aip5 assembly in different conditions, the authors should just choose a name for each situation, define it and use it throughout the whole paper to avoid confusion.

We appreciate reviewer #2 comments and suggestion. We agreed with reviewer #2 about using back the original title because it helps to convey the key message for Aip5 protein behavior and function. And we have also added one more sentence in the abstract as suggested to better illustrate the cellular response of Aip5 phase behavior *in vivo*. Furthermore, to avoid potential confusion, we changed the terms by using amorphous assembly for *in vitro* Aip5 assemblies and condensates/condensation for *in vivo* Aip5.